# BDDM: Bilateral Denoising Diffusion Models for Fast and High-Quality Speech Synthesis

**Max W. Y. Lam, Jun Wang, Dan Su**
Tencent AI Lab
Shenzhen, China
{maxwylam, joinerwang, dansu}@tencent.com

**Dong Yu**
Tencent AI Lab
Bellevue WA, USA
dyu@tencent.com

## Abstract

Diffusion probabilistic models (DPMs) and their extensions have emerged as competitive generative models yet confront challenges of efficient sampling. We propose a new bilateral denoising diffusion model (BDDM) that parameterizes both the forward and reverse processes with a schedule network and a score network, which can train with a novel bilateral modeling objective. We show that the new surrogate objective can achieve a lower bound of the log marginal likelihood tighter than a conventional surrogate. We also find that BDDM allows inheriting pre-trained score network parameters from any DPMs and consequently enables speedy and stable learning of the schedule network and optimization of a noise schedule for sampling. Our experiments demonstrate that BDDMs can generate high-fidelity audio samples with as few as three sampling steps. Moreover, compared to other state-of-the-art diffusion-based neural vocoders, BDDMs produce comparable or higher quality samples indistinguishable from human speech, notably with only seven sampling steps (143x faster than WaveGrad and 28.6x faster than DiffWave). We release our code at https://github.com/tencent-ailab/bddm.

## 1 Introduction

Deep generative models have shown a tremendous advancement in speech synthesis (van den Oord et al., 2016; Kalchbrenner et al., 2018; Prenger et al., 2019; Kumar et al., 2019; Kong et al., 2020b; Chen et al., 2020; Kong et al., 2021). Successful generative models can be mainly divided into two categories: generative adversarial network (GAN) (Goodfellow et al., 2014) based and likelihood-based. The former is based on adversarial learning, where the objective is to generate data indistinguishable from the training data. Yet, the training GANs can be very unstable, and the relevant training objectives are not suitable to compare against different GANs. The latter uses log-likelihood or surrogate objectives for training, but they also have intrinsic limitations regarding generation speed or quality. For example, the autoregressive models (van den Oord et al., 2016; Kalchbrenner et al., 2018), while being capable of generating high-fidelity data, are limited by their inherently slow sampling process and the poor scaling properties on high-dimensional data. Likewise, the flow-based models (Dinh et al., 2016; Kingma & Dhariwal, 2018; Chen et al., 2018; Papamakarios et al., 2021) rely on specialized architectures to build a normalized probability model, whose training is less parameter-efficient. Other prior works use surrogate objectives, such as the evidence lower bound in variational auto-encoders (Kingma & Welling, 2014; Rezende et al., 2014; Maaløe et al., 2019) and the contrastive divergence in energy-based models (Hinton, 2002; Carreira-Perpinan & Hinton, 2005). These models, despite showing improved speed, typically only work well for low-dimensional data, and, in general, the sample qualities are not competitive to the GAN-based and the autoregressive models (Bond-Taylor et al., 2021).

An up-and-coming class of likelihood-based models is the diffusion probabilistic models (DPMs) (Sohl-Dickstein et al., 2015), which introduces the idea of using a forward diffusion process to sequentially corrupt a given distribution and learning the reversal of such diffusion process to restore the data distribution for sampling. From a similar perspective, Song & Ermon (2019) proposed the score-based generative models by applying the score matching technique (Hyvarinen & Dayan,

2005) to train a neural network such that samples can be generated via Langevin dynamics. Along these two lines of research, Ho et al. (2020) proposed the denoising diffusion probabilistic models (DDPMs) for high-quality image syntheses. Dhariwal & Nichol (2021) demonstrated that improved DDPMs Nichol & Dhariwal (2021) are capable of generating high-quality images of comparable or even superior quality to the state-of-the-art (SOTA) GAN-based models. For speech syntheses, DDPMs were also applied in Wavegrad (Chen et al., 2020) and DiffWave (Kong et al., 2021) to produce higher-fidelity audio samples than the conventional non-autoregressive models (Yamamoto et al., 2020; Kumar et al., 2019; Yang et al., 2021; Bińkowski et al., 2020) and matched the quality of the SOTA autoregressive methods (Chen et al., 2020).

Despite the compelling results, the diffusion generative models are two to three orders of magnitude slower than other generative models such as GANs and VAEs. Their primary limitation is that they require up to thousands of diffusion steps during training to learn the target distribution. Therefore a large number of reverse steps are often required at sampling time. Recently, extensive investigations have been conducted to reduce the sampling steps for efficiently generating high-quality samples, which we will discuss in the related work in Section 2. Distinctively, we conceived that we might train a neural network to efficiently and adaptively estimate a much shorter noise schedule for sampling while achieving generation performances comparable or superior to the conventional DPMs. With such an incentive, after introducing the conventional DPMs as our background in Section 3, we propose in Section 4 bilateral denoising diffusion models (BDDMs), named after a bilateral modeling perspective – parameterizing the forward and reverse processes with a schedule network and a score network, respectively. We theoretically derive that the schedule network should be trained after the score network is optimized. For training the schedule network, we propose a novel objective to minimize the gap between a newly derived lower bound and the log marginal likelihood. We describe the training algorithm as well as the fast and high-quality sampling algorithm in Section 5. The training of the schedule network converges very fast using our newly derived objective, and its training only adds negligible overhead to DDPM's. In Section 6, our neural vocoding experiments demonstrated that BDDMs could generate high-fidelity samples with as few as three sampling steps. Moreover, our method can produce speech samples indistinguishable from human speech with only seven sampling steps (143x faster than WaveGrad and 28.6x faster than DiffWave).

## 2 RELATED WORK

Prior works showed that noise scheduling is crucial for efficient and high-fidelity data generation in DPMs. DDPMs (Ho et al., 2020) used a shared linear noise schedule for both training and sampling, which, however, requires thousands of sampling iterations to obtain competitive results. To speed up the sampling process, one class of related work, including (Chen et al., 2020; Kong et al., 2021; Nichol & Dhariwal, 2021), attempts to use a different, shorter noise schedule for sampling. For clarity, we thereafter denote the training noise schedule as $\boldsymbol{\beta} \in \mathbb{R}^T$ and the sampling noise schedule as $\hat{\boldsymbol{\beta}} \in \mathbb{R}^N$ with $N < T$. In particular, Chen et al. (2020) applied a grid search (GS) algorithm to select $\hat{\boldsymbol{\beta}}$. Unfortunately, GS becomes prohibitively slow when $N$ grows large, e.g., $N = 6$ took more than a day on a single NVIDIA Tesla P40 GPU. This is because the time costs of GS algorithm grow exponentially with $N$, i.e., $\mathcal{O}(9^N)$ with 9 bins as the default setting in Chen et al. (2020). Instead of searching, Kong et al. (2021) devised a fast sampling (FS) algorithm based on an expert-defined 6-step noise schedule for their score network. However, this specifically tuned noise schedule is hard to generalize to other score networks, tasks, or datasets.

Another class of noise scheduling methods searches for a subsequence of time indices of the training noise schedule, which we call the *time schedule*. DDIMs (Song et al., 2021) introduced an accelerated reverse process that relies on a pre-specified time schedule. A linear and a quadratic time schedule were used in DDIMs and showed superior generation quality over DDPMs within 10 to 100 sampling steps. Nichol & Dhariwal (2021) proposed a re-scaled noise schedule for fast sampling, but this also requires pre-specifying the time schedule and the training noise schedule. Nichol & Dhariwal (2021) also proposed learning variances for the reverse processes, whereas the variances of the forward processes, i.e., the noise schedule, which affected both the means and variances of the reverse processes, were not learnable. According to the results of (Song et al., 2021; Nichol & Dhariwal, 2021), using a linear or quadratic time schedule resulted in quite different performances in different datasets, implying that the optimal choice of schedule varies with the datasets. So, there remains a challenge in finding a short and effective schedule for fast sampling on different datasets.

Notably, Kong & Ping (2021) proposed a method to map a noise schedule to a time schedule for fast sampling. In this sense, searching for a time schedule becomes a sub-set of the noise scheduling problem, which resembles the above category of methods.

Although DPMs (Sohl-Dickstein et al., 2015) and DDPMs (Ho et al., 2019) mentioned that the noise schedule could be learned by re-parameterization, the approach was not investigated in their works. Closely related works that learn a noise schedule emerged until very recently. San-Roman et al. (2021) proposed a noise estimation (NE) method, which trained a neural net with a regression loss to estimate the noise scale from the noisy sample at each time point, and then predicted the next noise scale. However, NE requires a prior assumption of the noise schedule following a linear or Fibonacci rule. Most recently, a concurrent work to ours by Kingma et al. (2021) jointly trained a neural net to predict the signal-to-noise ratio (SNR) by maximizing the variational lower bound. The SNR was then used for noise scheduling. Different from ours, this scheduling neural net only took $t$ as input and is independent of the noisy sample generated during the loop of sampling process. Intrinsically, with limited information about the sampled data, the predicted SNR could deviate from the actual SNR of the noisy data during sampling.

# 3 BACKGROUND

## 3.1 DIFFUSION PROBABILISTIC MODELS (DPMS)

Given i.i.d. samples $\{\boldsymbol{x}_0 \in \mathbb{R}^D\}$ from an unknown data distribution $p_{\text{data}}(\boldsymbol{x}_0)$, diffusion probabilistic models (DPMs) (Sohl-Dickstein et al., 2015) define a forward process $q(\boldsymbol{x}_{1:T}|\boldsymbol{x}_0) = \prod_{t=1}^{T} q(\boldsymbol{x}_t|\boldsymbol{x}_{t-1})$ that converts any complex data distribution into a simple, tractable distribution after $T$ steps of diffusion. A reverse process $p_\theta(\boldsymbol{x}_{t-1}|\boldsymbol{x}_t)$ parameterized by $\theta$ is used to model the data distribution: $p_\theta(\boldsymbol{x}_0) = \int \pi(\boldsymbol{x}_T) \prod_{t=1}^{T} p_\theta(\boldsymbol{x}_{t-1}|\boldsymbol{x}_t) d\boldsymbol{x}_{1:T}$, where $\pi(\boldsymbol{x}_T)$ is the prior distribution for starting the reverse process. Then, the variational parameters $\theta$ can be learned by maximizing the standard log evidence lower bound (ELBO):

$$\mathcal{F}_{\text{elbo}} := \mathbb{E}_q \left[ \log p_\theta(\boldsymbol{x}_0|\boldsymbol{x}_1) - \sum_{t=2}^{T} D_{\text{KL}} \left( q(\boldsymbol{x}_{t-1}|\boldsymbol{x}_t, \boldsymbol{x}_0) || p_\theta(\boldsymbol{x}_{t-1}|\boldsymbol{x}_t) \right) - D_{\text{KL}} \left( q(\boldsymbol{x}_T|\boldsymbol{x}_0) || \pi(\boldsymbol{x}_T) \right) \right].$$
(1)

## 3.2 DENOISING DIFFUSION PROBABILISTIC MODELS (DDPMS)

As an extension to DPMs, denoising diffusion probabilistic models (DDPMs) (Ho et al., 2020) applied the score matching technique (Hyvarinen & Dayan, 2005; Song & Ermon, 2019) to define the reverse process. In particular, DDPMs considered a Gaussian diffusion process parameterized by a *noise schedule* $\boldsymbol{\beta} \in \mathbb{R}^T$ with $0 < \beta_1, \ldots, \beta_T < 1$:

$$q_{\boldsymbol{\beta}}(\boldsymbol{x}_{1:T}|\boldsymbol{x}_0) := \prod_{t=1}^{T} q_{\beta_t}(\boldsymbol{x}_t|\boldsymbol{x}_{t-1}), \quad \text{where} \quad q_{\beta_t}(\boldsymbol{x}_t|\boldsymbol{x}_{t-1}) := \mathcal{N}(\sqrt{1-\beta_t}\boldsymbol{x}_{t-1}, \beta_t \boldsymbol{I}). \quad (2)$$

Based on the nice property of isotropic Gaussians, one can express $\boldsymbol{x}_t$ directly conditioned on $\boldsymbol{x}_0$:

$$q_{\boldsymbol{\beta}}(\boldsymbol{x}_t|\boldsymbol{x}_0) = \mathcal{N}(\alpha_t \boldsymbol{x}_0, (1-\alpha_t^2)\boldsymbol{I}), \quad \text{where} \quad \alpha_t = \prod_{i=1}^{t} \sqrt{1-\beta_i}. \quad (3)$$

To revert this forward process, DDPMs employ a score network[1] $\boldsymbol{\epsilon}_\theta(\boldsymbol{x}_t, \alpha_t)$ to define

$$p_\theta(\boldsymbol{x}_{t-1}|\boldsymbol{x}_t) := \mathcal{N}\left( \frac{1}{\sqrt{1-\beta_t}} \left( \boldsymbol{x}_t - \frac{\beta_t}{\sqrt{1-\alpha_t^2}} \boldsymbol{\epsilon}_\theta(\boldsymbol{x}_t, \alpha_t) \right), \Sigma_t \right), \quad (4)$$

---

[1]Here, $\boldsymbol{\epsilon}_\theta(\boldsymbol{x}_t, \alpha_t)$ is conditioned on the continuous noise scale $\alpha_t$, as in (Song et al., 2020b; Chen et al., 2020). Alternatively, the score network can also be conditioned on a discrete time index $\boldsymbol{\epsilon}_\theta(\boldsymbol{x}_t, t)$, as in (Song et al., 2021; Ho et al., 2020). An approximate mapping of a noise schedule to a time schedule (Kong & Ping, 2021) exists, therefore we consider conditioning on noise scales as the general case.

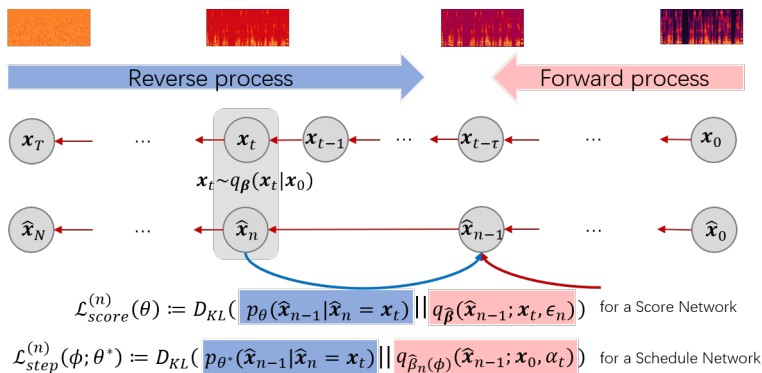

Figure 1: A bilateral denoising diffusion model (BDDM) introduces a *junctional* variable $\boldsymbol{x}_t$ and a schedule network $\phi$. The schedule network can optimize the shortened noise schedule $\hat{\beta}_n(\phi)$ if we know the score of the distribution at the junctional step, using the KL divergence to directly compare $p_{\theta^*}(\hat{\boldsymbol{x}}_{n-1}|\hat{\boldsymbol{x}}_n = \boldsymbol{x}_t)$ against the re-parameterized forward process posteriors.

where $\Sigma_t$ is the co-variance matrix defined for the reverse process. Ho et al. (2020) showed that setting $\Sigma_t = \tilde{\beta}_t \boldsymbol{I} = \frac{1-\alpha_{t-1}^2}{1-\alpha_t^2}\beta_t \boldsymbol{I}$ is optimal for a deterministic $\boldsymbol{x}_0$, while setting $\Sigma_t = \beta_t \boldsymbol{I}$ is optimal for a white noise $\boldsymbol{x}_0 \sim \mathcal{N}(\mathbf{0}, \boldsymbol{I})$. Alternatively, Nichol & Dhariwal (2021) proposed learnable variances by interpolating the two optimals with a jointly trained neural network, i.e., $\Sigma_{t,\theta}(\boldsymbol{x}) := \operatorname{diag}(\exp(\boldsymbol{v}_\theta(\boldsymbol{x})\log\beta_t + (1-\boldsymbol{v}_\theta(\boldsymbol{x}))\log\tilde{\beta}_t))$, where $\boldsymbol{v}_\theta(\boldsymbol{x}) \in \mathbb{R}^D$ is a trainable network.

Note that the calculation of the complete ELBO in Eq. (1) requires $T$ forward passes of the score network, which would make the training computationally prohibitive for a large $T$. To feasibly train the score network, instead of computing the complete ELBO, Ho et al. (2020) proposed an efficient training mechanism by sampling from a discrete uniform distribution: $t \sim \mathcal{U}\{1,...,T\}$, $\boldsymbol{x}_0 \sim p_{\text{data}}(\boldsymbol{x}_0)$, $\boldsymbol{\epsilon}_t \sim \mathcal{N}(\mathbf{0}, \boldsymbol{I})$ at each training iteration to compute the training loss:

$$\mathcal{L}_{\text{ddpm}}^{(t)}(\theta) := \left\| \boldsymbol{\epsilon}_t - \boldsymbol{\epsilon}_\theta\left(\alpha_t \boldsymbol{x}_0 + \sqrt{1-\alpha_t^2}\boldsymbol{\epsilon}_t, \alpha_t\right) \right\|_2^2, \tag{5}$$

which is a re-weighted form of $D_{\text{KL}}(q_{\boldsymbol{\beta}}(\boldsymbol{x}_{t-1}|\boldsymbol{x}_t, \boldsymbol{x}_0)||p_\theta(\boldsymbol{x}_{t-1}|\boldsymbol{x}_t))$. Ho et al. (2020) reported that the re-weighting worked effectively for learning $\theta$. Yet, we demonstrate it is deficient for learning the noise schedule $\boldsymbol{\beta}$ in our ablation experiment in Section 6.2.

## 4 BILATERAL DENOISING DIFFUSION MODELS (BDDMS)

### 4.1 PROBLEM FORMULATION

For fast sampling with DPMs, we strive for a noise schedule $\hat{\boldsymbol{\beta}}$ for sampling that is much shorter than the noise schedule $\boldsymbol{\beta}$ for training. As shown in Fig. 1, we define two separate diffusion processes corresponding to the noise schedules, $\boldsymbol{\beta}$ and $\hat{\boldsymbol{\beta}}$, respectively. The upper diffusion process parameterized by $\boldsymbol{\beta}$ is the same as in Eq. (2), whereas the lower process is defined as $q_{\hat{\boldsymbol{\beta}}}(\hat{\boldsymbol{x}}_{1:N}|\hat{\boldsymbol{x}}_0) = \prod_{n=1}^N q_{\hat{\beta}_n}(\hat{\boldsymbol{x}}_n|\hat{\boldsymbol{x}}_{n-1})$ with much fewer diffusion steps ($N \ll T$). In our problem formulation, $\boldsymbol{\beta}$ is given, but $\hat{\boldsymbol{\beta}}$ is unknown. The goal is to find a $\hat{\boldsymbol{\beta}}$ for the reverse process $p_\theta(\hat{\boldsymbol{x}}_{n-1}|\hat{\boldsymbol{x}}_n; \hat{\beta}_n)$ such that $\hat{\boldsymbol{x}}_0$ can be effectively recovered from $\hat{\boldsymbol{x}}_N$ with $N$ reverse steps.

### 4.2 MODEL DESCRIPTION

Although many prior arts (Ho et al., 2020; Chen et al., 2020; Song et al., 2021; San-Roman et al., 2021) directly applied a shortened linear or Fibonacci noise schedule to the reverse process, we argue that these are sub-optimal solutions. Theoretically, the diffusion process specified by a new shortened noise schedule is essentially different from the one used to train the score network $\theta$. Therefore, $\theta$ is not guaranteed suitable for reverting the shortened diffusion process. This issue motivated a novel modeling perspective to establish a link between the shortened schedule $\hat{\boldsymbol{\beta}}$ and the score network $\theta$, i.e., to have $\hat{\boldsymbol{\beta}}$ optimized according to $\theta$.

As a starting point, we consider an $N = \lfloor T/\tau \rfloor$, where $1 \leq \tau < T$ is a hyperparameter controlling the step size such that each diffusion step between two consecutive variables in the shorter diffusion process corresponds to $\tau$ diffusion steps in the longer one. Based on Eq. (2), we define the following:

$$q_{\hat{\beta}_{n+1}}(\hat{\boldsymbol{x}}_{n+1}|\hat{\boldsymbol{x}}_n = \boldsymbol{x}_t) := q_{\boldsymbol{\beta}}(\boldsymbol{x}_{t+\tau}|\boldsymbol{x}_t) = \mathcal{N}\left(\sqrt{\frac{\alpha_{t+\tau}^2}{\alpha_t^2}}\boldsymbol{x}_t, \left(1 - \frac{\alpha_{t+\tau}^2}{\alpha_t^2}\right)\boldsymbol{I}\right), \tag{6}$$

where $\boldsymbol{x}_t$ is an intermediate diffused variable we introduced to link the two differently indexed diffusion sequences. We call it a *junctional* variable, which can be easily generated given $\boldsymbol{x}_0$ and $\boldsymbol{\beta}$ during training: $\boldsymbol{x}_t = \alpha_t\boldsymbol{x}_0 + \sqrt{1 - \alpha_t^2}\boldsymbol{\epsilon}_n$.

Unfortunately, for the reverse process when $\boldsymbol{x}_0$ is not given, the junctional variable is intractable. However, our key observation is that while using the score by a score network $\theta^*$ trained for the long $\boldsymbol{\beta}$-parameterized diffusion process, a short noise schedule $\hat{\boldsymbol{\beta}}(\phi)$ can be optimized accordingly by introducing a schedule network $\phi$. We provide its mathematical derivations in Appendix A.3. Next, we present a formal definition of BDDM and derive its training objectives, $\mathcal{L}_{\text{score}}^{(n)}(\theta)$ and $\mathcal{L}_{\text{step}}^{(n)}(\phi; \theta^*)$, for the score network and the schedule network, respectively, in more detail.

## 4.3 SCORE NETWORK

Recall that a DDPM starts the reverse process with a white noise $\boldsymbol{x}_T \sim \mathcal{N}(\boldsymbol{0}, \boldsymbol{I})$ and takes $T$ steps to recover the data distribution:

$$p_\theta(\boldsymbol{x}_0) \stackrel{\text{DDPM}}{:=} \mathbb{E}_{\mathcal{N}(\boldsymbol{0},\boldsymbol{I})}\left[\mathbb{E}_{p_\theta(\boldsymbol{x}_{1:T-1}|\boldsymbol{x}_T)}\left[p_\theta(\boldsymbol{x}_0|\boldsymbol{x}_{1:T})\right]\right]. \tag{7}$$

A BDDM, in contrast, starts from the *junctional* variable $\boldsymbol{x}_t$, and reverts a shorter sequence of diffusion random variables with only $n$ steps:

$$p_\theta(\hat{\boldsymbol{x}}_0) \stackrel{\text{BDDM}}{:=} \mathbb{E}_{q_{\hat{\beta}}(\hat{\boldsymbol{x}}_{n-1};\boldsymbol{x}_t,\boldsymbol{\epsilon}_n)}\left[\mathbb{E}_{p_\theta(\hat{\boldsymbol{x}}_{1:n-2}|\hat{\boldsymbol{x}}_{n-1})}\left[p_\theta(\hat{\boldsymbol{x}}_0|\hat{\boldsymbol{x}}_{1:n-1})\right]\right], \quad 2 \leq n \leq N, \tag{8}$$

where $q_{\hat{\beta}}(\hat{\boldsymbol{x}}_{n-1}; \boldsymbol{x}_t, \boldsymbol{\epsilon}_n)$ is defined as a re-parameterization on the posterior:

$$q_{\hat{\boldsymbol{\beta}}}(\hat{\boldsymbol{x}}_{n-1}; \boldsymbol{x}_t, \boldsymbol{\epsilon}_n) := q_{\hat{\boldsymbol{\beta}}}\left(\hat{\boldsymbol{x}}_{n-1} \middle| \hat{\boldsymbol{x}}_n = \boldsymbol{x}_t, \hat{\boldsymbol{x}}_0 = \frac{\boldsymbol{x}_t - \sqrt{1 - \hat{\alpha}_n^2}\boldsymbol{\epsilon}_n}{\hat{\alpha}_n}\right) \tag{9}$$

$$= \mathcal{N}\left(\frac{1}{\sqrt{1 - \hat{\beta}_n}}\boldsymbol{x}_t - \frac{\hat{\beta}_n}{\sqrt{(1 - \hat{\beta}_n)(1 - \hat{\alpha}_n^2)}}\boldsymbol{\epsilon}_n, \frac{1 - \hat{\alpha}_{n-1}^2}{1 - \hat{\alpha}_n^2}\hat{\beta}_n\boldsymbol{I}\right), \tag{10}$$

where $\hat{\alpha}_n = \prod_{i=1}^n \sqrt{1 - \hat{\beta}_i}$, $\boldsymbol{x}_t = \alpha_t\boldsymbol{x}_0 + \sqrt{1 - \alpha_t^2}\boldsymbol{\epsilon}_n$ is the *junctional* variable that maps $\boldsymbol{x}_t$ to $\hat{\boldsymbol{x}}_n$ given an approximate index $t \sim \mathcal{U}\{(n-1)\tau, ..., n\tau - 1, n\tau\}$ and a sampled white noise $\boldsymbol{\epsilon}_n \sim \mathcal{N}(\boldsymbol{0}, \boldsymbol{I})$. Detailed derivation from Eq. (9) to (10) is provided in Appendix A.2.

### 4.3.1 TRAINING OBJECTIVE FOR SCORE NETWORK

With the above definition, a new form of lower bound to the log marginal likelihood can be derived such that $\log p_\theta(\hat{\boldsymbol{x}}_0) \geq \mathcal{F}_{\text{score}}^{(n)}(\theta) := -\mathcal{L}_{\text{score}}^{(n)}(\theta) - \mathcal{R}_\theta(\hat{\boldsymbol{x}}_0, \boldsymbol{x}_t)$, where

$$\mathcal{L}_{\text{score}}^{(n)}(\theta) := D_{\text{KL}}\left(p_\theta(\hat{\boldsymbol{x}}_{n-1}|\hat{\boldsymbol{x}}_n = \boldsymbol{x}_t) || q_{\hat{\beta}}(\hat{\boldsymbol{x}}_{n-1}; \boldsymbol{x}_t, \boldsymbol{\epsilon}_n)\right), \tag{11}$$

$$\mathcal{R}_\theta(\hat{\boldsymbol{x}}_0, \boldsymbol{x}_t) := -\mathbb{E}_{p_\theta(\hat{\boldsymbol{x}}_1|\hat{\boldsymbol{x}}_n = \boldsymbol{x}_t)}\left[\log p_\theta(\hat{\boldsymbol{x}}_0|\hat{\boldsymbol{x}}_1)\right]. \tag{12}$$

See detailed derivation in Proposition 1 in Appendix A.2. In the following Proposition 2, we prove that via the junctional variable $\boldsymbol{x}_t$, the solution $\theta^*$ for optimizing the objective $\mathcal{L}_{\text{ddpm}}^{(t)}(\theta), \forall t \in \{1, ..., T\}$ is also the solution for optimizing $\mathcal{L}_{\text{score}}^{(n)}(\theta), \forall n \in \{2, ..., N\}$. Thereby, we show that the score network $\theta$ can be trained with $\mathcal{L}_{\text{ddpm}}^{(t)}(\theta)$ and re-used for reverting the short diffusion process over $\hat{\boldsymbol{x}}_{N:0}$. Although the newly derived lower bound result in the same objective as the conventional score network, it for the first time establishes a link between the score network $\theta$ and $\hat{\boldsymbol{x}}_{N:0}$. The connection is essential for learning $\hat{\boldsymbol{\beta}}$, which we will describe next.

### 4.4 SCHEDULE NETWORK

In BDDMs, a schedule network is introduced to the forward process by re-parameterizing $\hat{\beta}_n$ as $\hat{\beta}_n(\phi) = f_\phi\left(\boldsymbol{x}_t; \hat{\beta}_{n+1}\right)$, and recall that during training, we can use $\boldsymbol{x}_t = \alpha_t \boldsymbol{x}_0 + \sqrt{1 - \alpha_t^2}\boldsymbol{\epsilon}_n$ and $\hat{\beta}_{n+1} = 1 - \frac{\alpha_{t+\tau}^2}{\alpha_t^2}$. Through the re-parameterization, the task of noise scheduling, i.e., searching for $\hat{\boldsymbol{\beta}}$, can now be reformulated as training a schedule network $f_\phi$ that ancestrally estimates data-dependent variances. The schedule network learns to predict $\hat{\beta}_n$ based on the current noisy sample $\boldsymbol{x}_t$ – this makes our method fundamentally different from existing and concurrent work, including Kingma et al. (2021) – as we reveal that, aside from $\hat{\beta}_{n+1}$, $t$, or $n$ that reflects diffusion step information, $\boldsymbol{x}_t$ is also essential for noise scheduling from a reverse direction at inference time.

Specifically, we adopt the ancestral step information ($\hat{\beta}_{n+1}$) to derive an upper bound for the current step while leaving the schedule network only to take the current noisy sample $\boldsymbol{x}_t$ as input to predict a relative change of noise scales against the ancestral step. First, we derive an upper bound of $\hat{\beta}_n$ by proving $0 < \hat{\beta}_n < \min\left\{1 - \frac{\hat{\alpha}_{n+1}^2}{1 - \hat{\beta}_{n+1}}, \hat{\beta}_{n+1}\right\}$ in Appendix A.1. Then, by multiplying the upper bound by a ratio estimated by a neural network $\sigma_\phi : \mathbb{R}^D \mapsto (0, 1)$, we define

$$f_\phi(\boldsymbol{x}_t; \hat{\beta}_{n+1}) := \min\left\{1 - \frac{\hat{\alpha}_{n+1}^2}{1 - \hat{\beta}_{n+1}}, \hat{\beta}_{n+1}\right\}\sigma_\phi(\boldsymbol{x}_t), \tag{13}$$

where the network parameter set $\phi$ is learned to estimate the ratio between two consecutive noise scales ($\hat{\beta}_n$ and $\hat{\beta}_{n+1}$) from the current noisy input $\boldsymbol{x}_t$.

Finally, at inference time for noise scheduling, starting from a maximum reverse steps ($N$) and two hyperparameters ($\hat{\alpha}_N, \hat{\beta}_N$), we ancestrally predict the noise scale $\hat{\beta}_n(\phi) = f_\phi\left(\hat{\boldsymbol{x}}_n; \hat{\beta}_{n+1}\right)$, for $n$ from $N$ to $1$, and cumulatively update the product $\hat{\alpha}_n = \frac{\hat{\alpha}_{n+1}}{\sqrt{1 - \hat{\beta}_{n+1}}}$.

#### 4.4.1 TRAINING OBJECTIVE FOR SCHEDULE NETWORK

Here we describe how to learn the network parameters $\phi$ effectively. First, we demonstrated that $\phi$ should be trained after $\theta$ is well-optimized, referring to Proposition 3 in Appendix A.3. The Proposition also shows that we are minimizing the gap between the lower bound $\mathcal{F}_{\text{score}}^{(n)}(\theta^*)$ and $\log p_{\theta^*}(\hat{\boldsymbol{x}}_0)$, i.e., $\log p_{\theta^*}(\hat{\boldsymbol{x}}_0) - \mathcal{F}_{\text{score}}^{(n)}(\theta^*)$, by minimizing the following objective

$$\mathcal{L}_{\text{step}}^{(n)}(\phi; \theta^*) := D_{\text{KL}}\left(p_{\theta^*}(\hat{\boldsymbol{x}}_{n-1}|\hat{\boldsymbol{x}}_n = \boldsymbol{x}_t)||q_{\hat{\beta}_n(\phi)}(\hat{\boldsymbol{x}}_{n-1}; \boldsymbol{x}_0, \alpha_t)\right), \tag{14}$$

which is defined as a KL divergence to directly compare $p_{\theta^*}(\hat{\boldsymbol{x}}_{n-1}|\hat{\boldsymbol{x}}_n = \boldsymbol{x}_t)$ against the re-parameterized forward process posteriors, which are tractable when conditioned on the junctional noise scale $\alpha_t$ and $\boldsymbol{x}_0$.

The detailed derivation of Eq. (14) is also provided in the proof of Proposition 3 to get its concrete formulas as shown in Step (8-10) in Alg. 2.

## 5 ALGORITHMS: TRAINING, NOISE SCHEDULING, AND SAMPLING

### 5.1 TRAINING SCORE AND SCHEDULE NETWORKS

Following the theoretical result in Appendix A.3, $\theta$ should be optimized before learning $\phi$. Thereby first, to train the score network $\boldsymbol{\epsilon}_\theta$, we refer to the settings in (Ho et al., 2020; Chen et al., 2020; Song et al., 2021) to define $\boldsymbol{\beta}$ as a linear noise schedule: $\beta_t = \beta_{\text{start}} + \frac{t}{T}(\beta_{\text{end}} - \beta_{\text{start}})$, for $1 \leq t \leq T$, where $\beta_{\text{start}}$ and $\beta_{\text{end}}$ are two hyperparameter that specifies the start value and the end value. This results in Algorithm 1, which resembles the training algorithm in (Ho et al., 2020).

Next, based on the converged score network $\theta^*$, we train the schedule network $\phi$. We draw an $n \sim \mathcal{U}\{2, \ldots, N\}$ at each training step, and then draw a $t \sim \mathcal{U}\{(n-1)\tau, \ldots, n\tau\}$. These together can be re-formulated as directly drawing $t \sim \mathcal{U}\{\tau, \ldots, T - \tau\}$ for a finer-scale time step. Then, we

---

**Algorithm 1** Training Score Network ($\theta$)

1: Given $T, \{\beta_t\}_{t=1}^T$
2: $\{\alpha_t\}_{t=1}^T = \{\prod_{i=1}^t \sqrt{1-\beta_t}\}_{t=1}^T$
3: **repeat**
4:   $\boldsymbol{x}_0 \sim p_{\text{data}}(\boldsymbol{x}_0)$
5:   $t \sim \mathcal{U}\{1, \dots, T\}$
6:   $\boldsymbol{\epsilon}_t \sim \mathcal{N}(\boldsymbol{0}, \boldsymbol{I})$
7:   $\boldsymbol{x}_t = \alpha_t \boldsymbol{x}_0 + \sqrt{1-\alpha_t^2}\boldsymbol{\epsilon}_t$
8:   $\mathcal{L}_{\text{ddpm}}^{(t)} = \|\boldsymbol{\epsilon}_t - \boldsymbol{\epsilon}_\theta(\boldsymbol{x}_t, \alpha_t)\|_2^2$
9:   Take a gradient descent step on $\nabla_\theta \mathcal{L}_{\text{ddpm}}^{(t)}$
10: **until** converged

---

**Algorithm 3** Noise Scheduling

1: Given $\theta^*, \hat{\alpha}_N, \hat{\beta}_N, \boldsymbol{x}_N \sim \mathcal{N}(\boldsymbol{0}, \boldsymbol{I})$
2: **for** $n = N$ to $2$ **do**
3:   $\hat{\boldsymbol{x}}_{n-1} \sim p_{\theta^*}(\hat{\boldsymbol{x}}_{n-1}|\hat{\boldsymbol{x}}_n; \hat{\alpha}_n, \hat{\beta}_n)$
4:   $\hat{\alpha}_{n-1} = \frac{\hat{\alpha}_n}{\sqrt{1-\hat{\beta}_n}}$
5:   $\hat{\beta}_{n-1} = \min\{1 - \hat{\alpha}_{n-1}^2, \hat{\beta}_n\}\sigma_\phi(\hat{\boldsymbol{x}}_{n-1})$
6:   **if** $\hat{\beta}_{n-1} < \beta_1$ **then**
7:     **return** $\hat{\beta}_n, \dots, \hat{\beta}_N$
8:   **end if**
9: **end for**
10: **return** $\hat{\beta}_1, \dots, \hat{\beta}_N$

---

**Algorithm 2** Training Schedule Network ($\phi$)

1: Given $\theta^*, \tau, T, \{\alpha_t, \beta_t\}_{t=1}^T$
2: **repeat**
3:   $\boldsymbol{x}_0 \sim p_{\text{data}}(\boldsymbol{x}_0)$
4:   $t \sim \mathcal{U}\{\tau, \dots, T - \tau\}$
5:   $\delta_t = 1 - \alpha_t^2$
6:   $\boldsymbol{\epsilon}_n \sim \mathcal{N}(\boldsymbol{0}, \boldsymbol{I})$
7:   $\boldsymbol{x}_t = \alpha_t \boldsymbol{x}_0 + \sqrt{\delta_t}\boldsymbol{\epsilon}_n$
8:   $\hat{\beta}_n = \min\left\{\delta_t, 1 - \frac{\alpha_{t+\tau}^2}{\alpha_t^2}\right\}\sigma_\phi(\boldsymbol{x}_t)$
9:   $C = 4^{-1}\log(\delta_t/\hat{\beta}_n) + 2^{-1}D\left(\hat{\beta}_n/\delta_t - 1\right)$
10:   $\mathcal{L}_{\text{step}}^{(n)} = \frac{\delta_t}{2(\delta_t - \hat{\beta}_n)}\left\|\boldsymbol{\epsilon}_n - \frac{\hat{\beta}_n}{\delta_t}\boldsymbol{\epsilon}_{\theta^*}(\boldsymbol{x}_t, \alpha_t)\right\|_2^2 + C$
11:   Take a gradient descent step on $\nabla_\phi \mathcal{L}_{\text{step}}^{(n)}$
12: **until** converged

---

**Algorithm 4** Sampling

1: Given $\theta^*, \{\hat{\beta}_n\}_{n=1}^{N_s}, \hat{\boldsymbol{x}}_{N_s} \sim \mathcal{N}(\boldsymbol{0}, \boldsymbol{I})$
2: $\{\hat{\alpha}_n\}_{n=1}^{N_s} = \left\{\prod_{i=1}^n \sqrt{1-\hat{\beta}_n}\right\}_{n=1}^{N_s}$
3: **for** $n = N_s$ to $1$ **do**
4:   $\hat{\boldsymbol{x}}_{n-1} \sim p_{\theta^*}(\hat{\boldsymbol{x}}_{n-1}|\hat{\boldsymbol{x}}_n; \hat{\alpha}_n, \hat{\beta}_n)$
5: **end for**
6: **return** $\hat{\boldsymbol{x}}_0$

---

sequentially compute the variables needed for calculating $\mathcal{L}_{\text{step}}^{(n)}(\phi; \theta^*)$, as presented in Algorithm 2. We observed that, although a linear schedule is used to define $\boldsymbol{\beta}$, the noise schedule of $\hat{\boldsymbol{\beta}}$ predicted by $f_\phi$ is not limited to but rather different from a linear one.

## 5.2 Noise scheduling for fast and high-quality sampling

After the score network and the schedule network are trained, the inference procedure can divide into two phases: (1) the noise scheduling phase and (2) the sampling phase.

First, we run the noise scheduling process similarly to a sampling process with $N$ iterations maximum. Different from training, where $\alpha_t$ is forward-computed, $\hat{\alpha}_n$ is instead a backward-computed variable (from $N$ to 1) that may deviate from the forward one because $\{\hat{\beta}_i\}_i^{n-1}$ are unknown in the noise scheduling phase during inference. To start noise scheduling, we first set two hyperparameters: $\hat{\alpha}_N$ and $\hat{\beta}_N$. We use $\beta_1$, the smallest noise scale seen in training, as a threshold to early stop the noise scheduling process so that we can ignore small noise scales ($< \beta_1$) that were never seen by the score network. Overall, the noise scheduling process presents in Algorithm 3.

In practice, we apply a grid search algorithm of $M$ bins to Algorithm 3, which takes $\mathcal{O}(M^2)$ time, to find proper values for $(\hat{\alpha}_N, \hat{\beta}_N)$. We used $M = 9$ as in (Chen et al., 2020). The grid search for our noise scheduling algorithm can be evaluated on a small subset of the training samples. Empirically, even as few as 1 sample for evaluation works well in our algorithm. Finally, given the predicted noise schedule $\hat{\boldsymbol{\beta}} \in \mathbb{R}^{N_s}$, we generate samples with $N_s$ sampling steps, as shown in Algorithm 4.

## 6 Experiments

We conducted a series of experiments on neural vocoding tasks to evaluate the proposed BDDMs. First, we compared BDDMs against several strongest models that have been published: the mixture of logistics (MoL) WaveNet (Oord et al., 2018) implemented in (Yamamoto, 2020), the WaveGlow (Prenger et al., 2019) implemented in (Valle, 2020), the MelGAN (Kumar et al., 2019) implemented in (Kumar, 2019), the HiFi-GAN (Kong et al., 2020b) implemented in (Kong et al., 2020a) and the two most recently proposed diffusion-based vocoders, i.e., WaveGrad (Chen et al., 2020) and DiffWave (Kong et al., 2021), both re-implemented in our code. The hyperparameter settings of BDDMs and all these models are detailed in Appendix B.

In addition, we also compared BDDMs to a variety of scheduling and acceleration techniques applicable to DDPMs, including the grid search (GS) approach in WaveGrad, the fast sampling (FS)

Table 1: Comparison of neural vocoders in terms of MOS with 95% confidence intervals, real-time factor (RTF) and model size in megabytes (MB) for inference. The highest score and the scores that are not significantly different from the highest score (p-values $\geq 0.05$) are bold-faced.

| Neural Vocoder | MOS | RTF | Size |
|---|---|---|---|
| Ground-truth | $4.64 \pm 0.08$ | — | — |
| WaveNet (MoL) (Oord et al., 2018) | $3.52 \pm 0.16$ | 318.6 | 282MB |
| WaveGlow (Prenger et al., 2019) | $3.03 \pm 0.15$ | 0.0198 | 645MB |
| MelGAN (Kumar et al., 2019) | $3.48 \pm 0.14$ | 0.00396 | 17MB |
| HiFi-GAN (Kong et al., 2020b) | $4.33 \pm 0.12$ | 0.0134 | 54MB |
| WaveGrad - 1000 steps (Chen et al., 2020) | $4.36 \pm 0.13$ | 38.2 | 183MB |
| DiffWave - 200 steps (Kong et al., 2021) | $\mathbf{4.49 \pm 0.13}$ | 7.30 | 27MB |
| BDDM - 3 steps ($\hat{\alpha}_N = 0.68, \hat{\beta}_N = 0.53$) | $3.64 \pm 0.13$ | 0.110 | 27MB |
| BDDM - 7 steps ($\hat{\alpha}_N = 0.62, \hat{\beta}_N = 0.42$) | $\mathbf{4.43 \pm 0.11}$ | 0.256 | 27MB |
| BDDM - 12 steps ($\hat{\alpha}_N = 0.67, \hat{\beta}_N = 0.12$) | $\mathbf{4.48 \pm 0.12}$ | 0.438 | 27MB |

Table 2: Comparison of sampling acceleration methods with the same score network and the same number of steps. The highest score and the scores that are not significantly different from the highest score (p-values $\geq 0.05$) are bold-faced.

| Steps | Acceleration Method | STOI | PESQ | MOS |
|---|---|---|---|---|
| 3 | GS (Chen et al., 2020) | $\mathbf{0.965 \pm 0.009}$ | $\mathbf{3.66 \pm 0.20}$ | $\mathbf{3.61 \pm 0.12}$ |
| | FS (Kong et al., 2021) | $0.939 \pm 0.023$ | $3.09 \pm 0.23$ | $3.10 \pm 0.12$ |
| | DDIM (Song et al., 2021) | $0.943 \pm 0.015$ | $3.42 \pm 0.27$ | $3.25 \pm 0.13$ |
| | NE (San-Roman et al., 2021) | $\mathbf{0.966 \pm 0.010}$ | $\mathbf{3.62 \pm 0.18}$ | $3.55 \pm 0.12$ |
| | BDDM | $\mathbf{0.966 \pm 0.011}$ | $\mathbf{3.63 \pm 0.24}$ | $\mathbf{3.64 \pm 0.13}$ |
| 7 | FS (Kong et al., 2021) | $\mathbf{0.981 \pm 0.006}$ | $3.68 \pm 0.24$ | $3.70 \pm 0.14$ |
| | DDIM (Song et al., 2021) | $0.974 \pm 0.008$ | $3.85 \pm 0.12$ | $3.94 \pm 0.12$ |
| | NE (San-Roman et al., 2021) | $0.978 \pm 0.007$ | $3.75 \pm 0.18$ | $4.02 \pm 0.11$ |
| | BDDM | $\mathbf{0.983 \pm 0.006}$ | $\mathbf{3.96 \pm 0.09}$ | $\mathbf{4.43 \pm 0.11}$ |
| 12 | DDIM (Song et al., 2021) | $0.979 \pm 0.006$ | $3.90 \pm 0.10$ | $4.16 \pm 0.12$ |
| | NE (San-Roman et al., 2021) | $0.981 \pm 0.007$ | $3.82 \pm 0.13$ | $3.98 \pm 0.14$ |
| | BDDM | $\mathbf{0.987 \pm 0.006}$ | $\mathbf{3.98 \pm 0.12}$ | $\mathbf{4.48 \pm 0.12}$ |

approach based on a user-defined 6-step schedule in DiffWave, the DDIMs (Song et al., 2021) and a noise estimation (NE) approach (San-Roman et al., 2021). For fair and reproducible comparison with other models and approaches, we used the LJSpeech dataset (Ito & Johnson, 2017), which consists of 13,100 22kHz audio clips of a female speaker. All diffusion models were trained on the same training split as in (Chen et al., 2020). We also replicated the comparative experiment of neural vocoding using a multi-speaker VCTK dataset (Yamagishi et al., 2019) as presented in Appendix C and obtained a result consistent with that obtained from the LJSpeech dataset.

## 6.1 SAMPLING QUALITY IN OBJECTIVE AND SUBJECTIVE METRICS

To assess the quality of each generated audio sample, we used both objective and subjective measures for comparing different neural vocoders given the same ground-truth spectrogram $s$ as the condition, i.e., $\epsilon_\theta(x, s, \alpha_t)$. Specifically, we used two scale-invariant metrics: the perceptual evaluation of speech quality (PESQ) (Rix et al., 2001) and the short-time objective intelligibility (STOI) (Taal et al., 2010) to measure the noisiness and the distortion of the generated speech relative to the reference speech. Mean opinion score (MOS) was also used as a subjective metric for evaluating the naturalness of the generated speech. The assessment scheme of MOS is included in Appendix B.

In Table 1, we compared BDDMs against the state-of-the-art (SOTA) vocoders. To predict noise schedules with different sampling steps (3, 7, and 12), we set three pairs of $\{\hat{\alpha}_N, \hat{\beta}_N\}$ for BDDMs by running on Algorithm 3 a quick hyperparameter grid search, which is detailed in Appendix B. Among the 9 evaluated vocoders, only our proposed BDDMs with 7 and 12 steps and DiffWave with 200 steps showed no statistic-significant difference from the ground-truth in terms of MOS. Moreover, BDDMs significantly outspeeded DiffWave in terms of RTFs. Notably, previous diffusion-

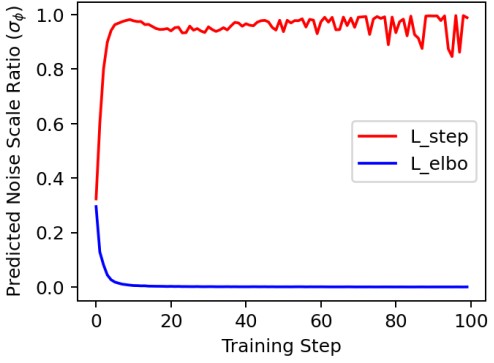

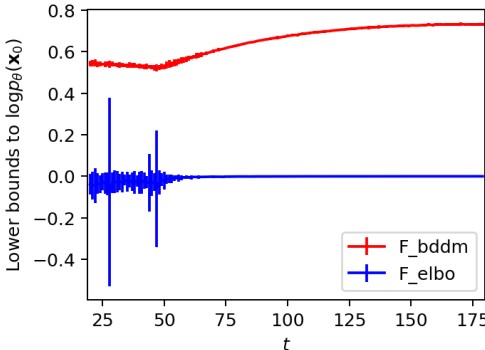

Figure 2: Different training losses for $\sigma_\phi$     Figure 3: Different lower bounds to $\log p_\theta(\boldsymbol{x}_0)$

based vocoders achieved high MOS scores at the cost of an unacceptable RTF for industrial deployment. In contrast, BDDMs managed to achieve a high standard of generation quality with only 7 sampling steps (143x faster than WaveGrad and 28.6x faster than DiffWave).

In Table 2, we evaluated BDDMs and alternative accelerated sampling methods, which used the same score network for a pair-to-pair comparison. The GS method performed stably when the step number was small (i.e., $N \leq 6$) but not scalable to more step numbers, which were therefore bypassed in the comparisons of 7 and 12 steps. The FS method by Song et al. (2021) was linearly interpolated to 3 and 7 steps for a fair comparison. Comparing its 7-step and 3-step results, we observed that the FS performance degraded drastically. Both the DDIM and the NE methods were stable across all the steps but were not performing competitively enough. In comparison, BDDMs consistently attained the leading scores across all the steps. This evaluation confirmed that BDDM was superior to other acceleration methods for DPMs in terms of both stability and quality.

## 6.2 Ablation Study and Analysis

We attribute the primary advantage of BDDMs to the newly derived objective $\mathcal{L}_{\text{step}}^{(n)}$ for learning $\phi$. To better reason about this, we performed an ablation study, where we substituted the proposed loss with the standard negative ELBO for learning $\phi$ as mentioned by Sohl-Dickstein et al. (2015). We plotted the network outputs with different training losses in Fig. 2. It turned out that, when using $\mathcal{L}_{\text{elbo}}^{(n)}$ to learn $\phi$, the network output rapidly collapsed to zero within several training steps; whereas, the network trained with $\mathcal{L}_{\text{step}}^{(n)}$ produced fluctuating outputs. The fluctuation is a desirable property showing the network properly predicts $t$-dependent noise scales, as $t$ is a random time step drawn from a uniform distribution in training.

By setting $\hat{\boldsymbol{\beta}} = \boldsymbol{\beta}$, we empirically validated that $\mathcal{F}_{\text{bddm}}^{(t)} := \mathcal{F}_{\text{score}}^{(t)} + \mathcal{L}_{\text{step}}^{(t)} \geq \mathcal{F}_{\text{elbo}}^{(t)}$ with their respective values at $t \in [20, 180]$ using the same optimized $\theta^*$. Each value is provided with 95% confidence intervals, as shown in Fig. 3. In this experiment, we used the LJ speech dataset and set $T = 200$ and $\tau = 20$. Notably, we dropped their common entropy term $\mathcal{R}_\theta(\hat{\boldsymbol{x}}_0, \boldsymbol{x}_t) < 0$ to mainly compare their KL divergences. This explains those positive lower bound values in the plot. The graph shows that our proposed bound $\mathcal{F}_{\text{bddm}}^{(t)}$ is always a tighter lower bound than the standard one across all examined $t$. Moreover, we found that $\mathcal{F}_{\text{bddm}}^{(t)}$ attained low values with a relatively much lower variance for $t \leq 50$, where $\mathcal{F}_{\text{elbo}}^{(t)}$ was highly volatile. This implies that $\mathcal{F}_{\text{bddm}}^{(t)}$ better tackles the difficult training part, i.e., when the score becomes more challenging to estimate as $t \to 0$.

## 7 Conclusions

BDDMs parameterize the forward and reverse processes with a schedule network and a score network, of which the former's optimization is tied with the latter by introducing a junctional variable. We derived a new lower bound that leads to the same training loss for the score network as in DDPMs (Ho et al., 2020), which thus enables inheriting any pre-trained score networks in DDPMs. We also showed that training the schedule network after a well-optimized score network can be viewed as tightening the lower bound. Followed from the theoretical results, an efficient training algorithm and a noise scheduling algorithm were respectively designed for BDDMs. Finally, in our experiments, BDDMs showed a clear edge over the previous diffusion-based vocoders.

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

## A    THEORETICAL DERIVATIONS FOR BDDMS

In this section, we provide the theoretical supports for the following:

- The derivation for upper bounding $\hat{\beta}_n$ (see Appendix A.1).
- The score network $\theta$ trained with $\mathcal{L}_{\text{ddpm}}^{(t)}(\theta)$ for the reverse process $p_\theta(\boldsymbol{x}_{t-1}|\boldsymbol{x}_t)$ can be re-used for the reverse process $p_\theta(\hat{\boldsymbol{x}}_{n-1}|\hat{\boldsymbol{x}}_n)$ (see Appendix A.2).
- The schedule network $\phi$ can be trained with $\mathcal{L}_{\text{step}}^{(n)}(\phi; \theta^*)$ after the score network $\theta$ is optimized. (see Appendix A.3).

### A.1    DERIVING AN UPPER BOUND FOR NOISE SCALE

Since monotonic noise schedules have been successfully applied to in many prior arts including DPMs (Ho et al., 2020; Kingma et al., 2021) and score-based methods (Song et al., 2020a; Song & Ermon, 2020), we also follow the monotonic assumption and derive an upper bound for $\hat{\beta}_n$ as below:

**Remark 1.** *Suppose the noise schedule for sampling is monotonic, i.e., $0 < \hat{\beta}_1 < \ldots < \hat{\beta}_N < 1$, then, for $1 \leq n < N$, $\hat{\beta}_n$ satisfies the following inequality:*

$$0 < \hat{\beta}_n < \min\left\{1 - \frac{\hat{\alpha}_{n+1}^2}{1 - \hat{\beta}_{n+1}}, \hat{\beta}_{n+1}\right\}. \tag{15}$$

*Proof.* By the general definition of noise schedule, we know that $0 < \hat{\beta}_1, \ldots, \hat{\beta}_N < 1$ (Note: no inequality sign in between). Given that $\hat{\alpha}_n = \prod_{i=1}^n \sqrt{1 - \hat{\beta}_i}$, we also have $0 < \hat{\alpha}_1, \ldots, \hat{\alpha}_t < 1$. First, we show that $\hat{\beta}_n < 1 - \frac{\hat{\alpha}_{n+1}^2}{1 - \hat{\beta}_{n+1}}$:

$$\hat{\alpha}_{n-1} = \frac{\hat{\alpha}_n}{\sqrt{1 - \hat{\beta}_n}} < 1 \iff \hat{\beta}_n < 1 - \hat{\alpha}_n^2 = 1 - \frac{\hat{\alpha}_{n+1}^2}{1 - \hat{\beta}_{n+1}}. \tag{16}$$

Next, we show that $\hat{\beta}_n < 1 - \hat{\alpha}_{n+1}$:

$$\frac{\hat{\alpha}_n}{\sqrt{1 - \hat{\beta}_n}} = \frac{\hat{\alpha}_n\sqrt{1 - \hat{\beta}_n}}{1 - \hat{\beta}_n} = \frac{\hat{\alpha}_{n+1}}{1 - \hat{\beta}_n} < 1 \iff \hat{\beta}_n < 1 - \hat{\alpha}_{n+1}. \tag{17}$$

Now, we have $\hat{\beta}_n < \min\left\{1 - \frac{\hat{\alpha}_{n+1}^2}{1 - \hat{\beta}_{n+1}}, 1 - \hat{\alpha}_{n+1}\right\}$. When $1 - \hat{\alpha}_{n+1} < 1 - \frac{\hat{\alpha}_{n+1}^2}{1 - \hat{\beta}_{n+1}}$, we can show that $\hat{\beta}_{n+1} < 1 - \hat{\alpha}_{n+1}$:

$$1 - \hat{\alpha}_{n+1} < 1 - \frac{\hat{\alpha}_{n+1}^2}{1 - \hat{\beta}_{n+1}} = 1 - \hat{\alpha}_n^2 \iff \hat{\alpha}_{n+1} > \hat{\alpha}_n^2 \iff \frac{\hat{\alpha}_{n+1}^2}{\hat{\alpha}_n^2} > \hat{\alpha}_{n+1} \tag{18}$$

$$\iff 1 - \frac{\hat{\alpha}_{n+1}^2}{\hat{\alpha}_n^2} < 1 - \hat{\alpha}_{n+1} \iff \hat{\beta}_{n+1} < 1 - \hat{\alpha}_{n+1}. \tag{19}$$

By the assumption of monotonic sequence, we also have $\hat{\beta}_n < \hat{\beta}_{n+1}$. Knowing that $\hat{\beta}_{n+1} < 1 - \hat{\alpha}_{n+1}$ is always true, we obtain a tighter bound for $\hat{\beta}_n$: $0 < \hat{\beta}_n < \min\left\{1 - \frac{\hat{\alpha}_{n+1}^2}{1 - \hat{\beta}_{n+1}}, \hat{\beta}_{n+1}\right\}$. □

### A.2    DERIVING THE TRAINING OBJECTIVE FOR SCORE NETWORK

First, followed from the data distribution modeling of BDDMs as proposed in Eq. (8):

$$p_\theta(\hat{\boldsymbol{x}}_0) := \mathbb{E}_{\hat{\boldsymbol{x}}_{n-1} \sim q_{\hat{\beta}}(\hat{\boldsymbol{x}}_{n-1}; \boldsymbol{x}_t, \boldsymbol{\epsilon}_n)} \left[ \mathbb{E}_{\hat{\boldsymbol{x}}_{1:n-2} \sim p_\theta(\hat{\boldsymbol{x}}_{1:n-2}|\hat{\boldsymbol{x}}_{n-1})} \left[ p_\theta(\hat{\boldsymbol{x}}_0|\hat{\boldsymbol{x}}_{1:n-1}) \right] \right], \tag{20}$$

we can derive a new lower bound to the log marginal likelihood as follows:

**Proposition 1.** *Given $\boldsymbol{x}_t \sim q_{\boldsymbol{\beta}}(\boldsymbol{x}_t|\boldsymbol{x}_0)$, the following lower bound holds for $n \in \{2, \ldots, N\}$:*

$$\log p_\theta(\hat{\boldsymbol{x}}_0) \geq \mathcal{F}_{score}^{(n)}(\theta) := -\mathcal{L}_{score}^{(n)}(\theta) - \mathcal{R}_\theta(\hat{\boldsymbol{x}}_0, \boldsymbol{x}_t), \tag{21}$$

*where*

$$\mathcal{L}_{score}^{(n)}(\theta) := D_{\mathrm{KL}}\left(p_\theta(\hat{\boldsymbol{x}}_{n-1}|\hat{\boldsymbol{x}}_n = \boldsymbol{x}_t)||q_{\hat{\boldsymbol{\beta}}}(\hat{\boldsymbol{x}}_{n-1}; \boldsymbol{x}_t, \boldsymbol{\epsilon}_n)\right), \tag{22}$$

$$\mathcal{R}_\theta(\hat{\boldsymbol{x}}_0, \boldsymbol{x}_t) := -\mathbb{E}_{p_\theta(\hat{\boldsymbol{x}}_1|\hat{\boldsymbol{x}}_n = \boldsymbol{x}_t)}\left[\log p_\theta(\hat{\boldsymbol{x}}_0|\hat{\boldsymbol{x}}_1)\right]. \tag{23}$$

*Proof.*

$$\log p_\theta(\hat{\boldsymbol{x}}_0) = \log \int p_\theta(\hat{\boldsymbol{x}}_{0:n-2}|\hat{\boldsymbol{x}}_{n-1})q_{\hat{\boldsymbol{\beta}}}(\hat{\boldsymbol{x}}_{n-1}; \boldsymbol{x}_t, \boldsymbol{\epsilon}_n)d\hat{\boldsymbol{x}}_{1:n-1} \tag{24}$$

$$= \log \int p_\theta(\hat{\boldsymbol{x}}_{0:n-2}|\hat{\boldsymbol{x}}_{n-1})q_{\hat{\boldsymbol{\beta}}}(\hat{\boldsymbol{x}}_{n-1}; \boldsymbol{x}_t, \boldsymbol{\epsilon}_n)\frac{p_\theta(\hat{\boldsymbol{x}}_{1:n-1}|\hat{\boldsymbol{x}}_n = \boldsymbol{x}_t)}{p_\theta(\hat{\boldsymbol{x}}_{1:n-1}|\hat{\boldsymbol{x}}_n = \boldsymbol{x}_t)}d\hat{\boldsymbol{x}}_{1:n-1} \tag{25}$$

$$= \log \mathbb{E}_{p_\theta(\hat{\boldsymbol{x}}_{1,n-1}|\hat{\boldsymbol{x}}_n = \boldsymbol{x}_t)}\left[\frac{p_\theta(\hat{\boldsymbol{x}}_0|\hat{\boldsymbol{x}}_1)q_{\hat{\boldsymbol{\beta}}}(\hat{\boldsymbol{x}}_{n-1}; \boldsymbol{x}_t, \boldsymbol{\epsilon}_n)}{p_\theta(\hat{\boldsymbol{x}}_{n-1}|\hat{\boldsymbol{x}}_n = \boldsymbol{x}_t)}\right] \tag{26}$$

$$[\text{Jensen's Inequality}] \geq \mathbb{E}_{p_\theta(\hat{\boldsymbol{x}}_1, \hat{\boldsymbol{x}}_{n-1}|\hat{\boldsymbol{x}}_n = \boldsymbol{x}_t)}\left[\log \frac{p_\theta(\hat{\boldsymbol{x}}_0|\hat{\boldsymbol{x}}_1)q_{\hat{\boldsymbol{\beta}}}(\hat{\boldsymbol{x}}_{n-1}; \boldsymbol{x}_t, \boldsymbol{\epsilon}_n)}{p_\theta(\hat{\boldsymbol{x}}_{n-1}|\hat{\boldsymbol{x}}_n = \boldsymbol{x}_t)}\right] \tag{27}$$

$$= \mathbb{E}_{p_\theta(\hat{\boldsymbol{x}}_1|\hat{\boldsymbol{x}}_n = \boldsymbol{x}_t)}\left[\log p_\theta(\hat{\boldsymbol{x}}_0|\hat{\boldsymbol{x}}_1)\right] - D_{\mathrm{KL}}\left(p_\theta(\hat{\boldsymbol{x}}_{n-1}|\hat{\boldsymbol{x}}_n = \boldsymbol{x}_t)||q_{\hat{\boldsymbol{\beta}}}(\hat{\boldsymbol{x}}_{n-1}; \boldsymbol{x}_t, \boldsymbol{\epsilon}_n)\right) \tag{28}$$

$$= -\mathcal{L}_{score}^{(n)}(\theta) - \mathcal{R}_\theta(\hat{\boldsymbol{x}}_0, \boldsymbol{x}_t) \tag{29}$$

$$\square$$

Next, we show that the score network $\theta$ trained with $\mathcal{L}_{ddpm}^{(t)}(\theta)$ can be re-used in BDDMs. We first provide the derivation for Eq. (9- 10). We have

$$q_{\hat{\boldsymbol{\beta}}}(\hat{\boldsymbol{x}}_{n-1}; \boldsymbol{x}_t, \boldsymbol{\epsilon}_n) := q_{\hat{\boldsymbol{\beta}}}\left(\hat{\boldsymbol{x}}_{n-1}|\hat{\boldsymbol{x}}_n = \boldsymbol{x}_t, \hat{\boldsymbol{x}}_0 = \frac{\boldsymbol{x}_t - \sqrt{1 - \hat{\alpha}_n^2}\boldsymbol{\epsilon}_n}{\hat{\alpha}_n}\right) \tag{30}$$

$$= \mathcal{N}\left(\frac{\hat{\alpha}_{n-1}\hat{\beta}_n}{1 - \hat{\alpha}_n^2}\frac{\boldsymbol{x}_t - \sqrt{1 - \hat{\alpha}_n^2}\boldsymbol{\epsilon}_n}{\hat{\alpha}_n} + \frac{\sqrt{1 - \hat{\beta}_n}(1 - \hat{\alpha}_{n-1}^2)}{1 - \hat{\alpha}_n^2}\boldsymbol{x}_t, \frac{1 - \hat{\alpha}_{n-1}^2}{1 - \hat{\alpha}_n^2}\hat{\beta}_n\boldsymbol{I}\right) \tag{31}$$

$$= \mathcal{N}\left(\left(\frac{\hat{\alpha}_{n-1}\hat{\beta}_n}{\hat{\alpha}_n(1 - \hat{\alpha}_n^2)} + \frac{\sqrt{1 - \hat{\beta}_n}(1 - \hat{\alpha}_{n-1}^2)}{1 - \hat{\alpha}_n^2}\right)\boldsymbol{x}_t - \frac{\hat{\alpha}_{n-1}\hat{\beta}_n}{\hat{\alpha}_n\sqrt{1 - \hat{\alpha}_n^2}}\boldsymbol{\epsilon}_n, \frac{1 - \hat{\alpha}_{n-1}^2}{1 - \hat{\alpha}_n^2}\hat{\beta}_n\boldsymbol{I}\right) \tag{32}$$

$$= \mathcal{N}\left(\frac{1}{\sqrt{1 - \hat{\beta}_n}}\boldsymbol{x}_t - \frac{\hat{\beta}_n}{\sqrt{(1 - \hat{\beta}_n)(1 - \hat{\alpha}_n^2)}}\boldsymbol{\epsilon}_n, \frac{1 - \hat{\alpha}_{n-1}^2}{1 - \hat{\alpha}_n^2}\hat{\beta}_n\boldsymbol{I}\right). \tag{33}$$

**Proposition 2.** *Suppose $\boldsymbol{x}_t \sim q_{\boldsymbol{\beta}}(\boldsymbol{x}_t|\boldsymbol{x}_0)$, then any solution satisfying $\theta^* = argmin_\theta \mathcal{L}_{ddpm}^{(t)}(\theta), \forall t \in \{1, ..., T\}$, also satisfies $\theta^* = argmin_\theta \mathcal{L}_{score}^{(n)}(\theta), \forall n \in \{2, ..., N\}$.*

*Proof.* By the definition in Eq. (4), we have

$$p_\theta(\hat{\boldsymbol{x}}_{n-1}|\hat{\boldsymbol{x}}_n = \boldsymbol{x}_t) = \mathcal{N}\left(\frac{1}{\sqrt{1 - \hat{\beta}_n}}\left(\boldsymbol{x}_t - \frac{\hat{\beta}_n}{\sqrt{1 - \hat{\alpha}_n^2}}\boldsymbol{\epsilon}_\theta(\boldsymbol{x}_t, \hat{\alpha}_n)\right), \frac{1 - \hat{\alpha}_{n-1}^2}{1 - \hat{\alpha}_n^2}\hat{\beta}_n\boldsymbol{I}\right). \tag{34}$$

Here, from the training objective in Eq. (5), since $\boldsymbol{x}_t = \alpha_t\boldsymbol{x}_0 + \sqrt{1 - \alpha_t^2}\boldsymbol{\epsilon}_n$, the noise scale argument for the score network is known to be $\alpha_t$. Therefore, we can use $\boldsymbol{\epsilon}_\theta(\boldsymbol{x}_t, \alpha_t)$ instead of $\boldsymbol{\epsilon}_\theta(\boldsymbol{x}_t, \hat{\alpha}_n)$ for

expanding $\mathcal{L}_{\text{score}}^{(n)}(\theta)$. Since $p_\theta(\hat{\boldsymbol{x}}_{n-1}|\hat{\boldsymbol{x}}_n = \boldsymbol{x}_t)$ and $q_{\hat{\boldsymbol{\beta}}}(\hat{\boldsymbol{x}}_{n-1}; \boldsymbol{x}_t, \boldsymbol{\epsilon}_n)$ are two isotropic Gaussians with the same variance, the KL divergence is a scaled $\ell$2-norm of their means' difference:

$$\mathcal{L}_{\text{score}}^{(n)}(\theta) := D_{\text{KL}}\left(p_\theta(\hat{\boldsymbol{x}}_{n-1}|\hat{\boldsymbol{x}}_n = \boldsymbol{x}_t)||q_{\hat{\boldsymbol{\beta}}}(\hat{\boldsymbol{x}}_{n-1}; \boldsymbol{x}_t, \boldsymbol{\epsilon}_n)\right) \tag{35}$$

$$= \frac{1 - \hat{\alpha}_n^2}{2(1 - \hat{\alpha}_{n-1}^2)\hat{\beta}_n}\left\|\frac{1}{\sqrt{1 - \hat{\beta}_n}}\left(\boldsymbol{x}_t - \frac{\hat{\beta}_n}{\sqrt{1 - \hat{\alpha}_n^2}}\boldsymbol{\epsilon}_\theta\left(\boldsymbol{x}_t, \alpha_t\right)\right)\right. \tag{36}$$

$$\left. - \left(\frac{1}{\sqrt{1 - \hat{\beta}_n}}\boldsymbol{x}_t - \frac{\hat{\beta}_n}{\sqrt{(1 - \hat{\beta}_n)(1 - \hat{\alpha}_n^2)}}\boldsymbol{\epsilon}_n\right)\right\|_2^2 \tag{37}$$

$$= \frac{(1 - \hat{\beta}_n)(1 - \hat{\alpha}_n^2)}{2(1 - \hat{\beta}_n - \hat{\alpha}_n^2)\hat{\beta}_n}\left\|\frac{\hat{\beta}_n}{\sqrt{(1 - \hat{\beta}_n)(1 - \hat{\alpha}_n^2)}}\left(\boldsymbol{\epsilon}_n - \boldsymbol{\epsilon}_\theta\left(\boldsymbol{x}_t, \alpha_t\right)\right)\right\|_2^2 \tag{38}$$

$$= \frac{(1 - \hat{\beta}_n)(1 - \hat{\alpha}_n^2)}{2(1 - \hat{\beta}_n - \hat{\alpha}_n^2)\hat{\beta}_n}\frac{\hat{\beta}_n^2}{(1 - \hat{\alpha}_n^2)(1 - \hat{\beta}_n)}\left\|\boldsymbol{\epsilon}_n - \boldsymbol{\epsilon}_\theta\left(\boldsymbol{x}_t, \alpha_t\right)\right\|_2^2 \tag{39}$$

$$= \frac{\hat{\beta}_n}{2(1 - \hat{\beta}_n - \hat{\alpha}_n^2)}\left\|\boldsymbol{\epsilon}_n - \boldsymbol{\epsilon}_\theta\left(\alpha_t\boldsymbol{x}_0 + \sqrt{1 - \alpha_t^2}\boldsymbol{\epsilon}_n, \alpha_t\right)\right\|_2^2, \tag{40}$$

which is proportional to $\mathcal{L}_{\text{ddpm}}^{(t)} := \left\|\boldsymbol{\epsilon}_n - \boldsymbol{\epsilon}_\theta\left(\alpha_t\boldsymbol{x}_0 + \sqrt{1 - \alpha_t^2}\boldsymbol{\epsilon}_n, \alpha_t\right)\right\|_2^2$ as defined in Eq. (5). Thus,

$$\text{argmin}_\theta\mathcal{L}_{\text{ddpm}}^{(t)}(\theta) \equiv \text{argmin}_\theta\mathcal{L}_{\text{score}}^{(n)}(\theta). \tag{41}$$

$$\square$$

Next, we can simplify $\mathcal{R}_\theta(\hat{\boldsymbol{x}}_0, \boldsymbol{x}_t)$ to a reconstruction loss for $\hat{\boldsymbol{x}}_0$:

$$\mathcal{R}_\theta(\hat{\boldsymbol{x}}_0, \boldsymbol{x}_t) := -\mathbb{E}_{p_\theta(\hat{\boldsymbol{x}}_1|\hat{\boldsymbol{x}}_n = \boldsymbol{x}_t)}\left[\log p_\theta(\hat{\boldsymbol{x}}_0|\hat{\boldsymbol{x}}_1)\right] \tag{42}$$

$$= \mathbb{E}_{p_\theta(\hat{\boldsymbol{x}}_1|\hat{\boldsymbol{x}}_n = \boldsymbol{x}_t)}\left[\log\mathcal{N}\left(\frac{1}{\sqrt{1 - \hat{\beta}_1}}\left(\hat{\boldsymbol{x}}_1 - \frac{\hat{\beta}_1}{\sqrt{1 - \hat{\alpha}_1^2}}\boldsymbol{\epsilon}_\theta(\hat{\boldsymbol{x}}_1, \hat{\alpha}_1), \hat{\beta}_1\mathbf{I}\right)\right)\right] \tag{43}$$

$$= \mathbb{E}_{p_\theta(\hat{\boldsymbol{x}}_1|\hat{\boldsymbol{x}}_n = \boldsymbol{x}_t)}\left[\frac{D}{2}\log 2\pi\hat{\beta}_1 + \frac{1}{2\hat{\beta}_1}\left\|\hat{\boldsymbol{x}}_0 - \frac{1}{\sqrt{1 - \hat{\beta}_1}}\left(\hat{\boldsymbol{x}}_1 - \frac{\hat{\beta}_1}{\sqrt{\hat{\beta}_1}}\boldsymbol{\epsilon}_\theta(\hat{\boldsymbol{x}}_1, \hat{\alpha}_1)\right)\right\|_2^2\right] \tag{44}$$

$$= \frac{D}{2}\log 2\pi\hat{\beta}_1 + \frac{1}{2\hat{\beta}_1}\mathbb{E}_{p_\theta(\hat{\boldsymbol{x}}_1|\hat{\boldsymbol{x}}_n = \boldsymbol{x}_t)}\left[\left\|\hat{\boldsymbol{x}}_0 - \frac{1}{\sqrt{1 - \hat{\beta}_1}}\left(\hat{\boldsymbol{x}}_1 - \sqrt{\hat{\beta}_1}\boldsymbol{\epsilon}_\theta(\hat{\boldsymbol{x}}_1, \hat{\alpha}_1)\right)\right\|_2^2\right], \tag{45}$$

where $p_\theta(\hat{\boldsymbol{x}}_1|\hat{\boldsymbol{x}}_n = \boldsymbol{x}_t)$ can be efficiently sampled using the reverse process in (Song et al., 2021). Yet, in practice, similar to the training in (Song et al., 2021; Chen et al., 2020; Kong et al., 2021), we dropped $\mathcal{R}_\theta(\hat{\boldsymbol{x}}_0, \boldsymbol{x}_t)$ when training $\theta$. In theory, we know that $\mathcal{R}_\theta(\hat{\boldsymbol{x}}_0, \boldsymbol{x}_t)$ achieves its optimal value at $\theta^* = \text{argmin}_\theta\|\boldsymbol{\epsilon}_\theta(\hat{\boldsymbol{x}}_1, \hat{\alpha}_1) - \boldsymbol{\epsilon}_1\|_2^2$, which shares a similar objective as $\mathcal{L}_{\text{ddpm}}^{(t)}$. By minimizing $\mathcal{L}_{\text{ddpm}}^{(t)}$, we train a score network $\theta^*$ that best minimizes $\Delta\boldsymbol{\epsilon}_t := \|\boldsymbol{\epsilon}_t - \boldsymbol{\epsilon}_{\theta^*}(\alpha_t\boldsymbol{x}_0 + \sqrt{1 - \alpha_t^2}\boldsymbol{\epsilon}_t, \alpha_t)\|_2^2$ for all $1 \leq t \leq T$. Since the first diffusion step has the smallest effect on corrupting $\hat{\boldsymbol{x}}_0$ (i.e., $\beta_1 \approx 0$), it suffices to consider a $\hat{\alpha}_1 = \sqrt{1 - \beta_1} = \alpha_1$, in which case we can jointly minimize $\mathcal{R}_\theta(\hat{\boldsymbol{x}}_0, \boldsymbol{x}_t)$ by minimizing $\mathcal{L}_{\text{ddpm}}^{(1)}$.

In this sense, during training, given $\boldsymbol{x}_t \sim q_{\boldsymbol{\beta}}(\boldsymbol{x}_t|\boldsymbol{x}_0)$, we can train the score network with the same training objective as in DDPMs and DDIMs. Practically, it is beneficial for BDDMs as we can re-use the score network $\theta$ of any well-trained DDPM or DDIM.

### A.3 DERIVING THE TRAINING OBJECTIVE FOR SCHEDULE NETWORK

Given that $\theta$ can be trained to maximize the log evidence with the pre-specified noise schedule $\boldsymbol{\beta}$ for training, the consequent question of interest in BDDMs is how to find a fast and good enough noise schedule $\hat{\boldsymbol{\beta}} \in \mathbb{R}^N$ for sampling given an optimized $\theta^*$. In BDDMs, this problem is reduced to how to effectively learn the network parameters $\phi$ .

**Proposition 3.** *Suppose $\theta$ has been optimized and hypothetically converged to the optimal $\theta^*$, where by optimal it means that with $\theta^*$ we have $p_{\theta^*}(\hat{\boldsymbol{x}}_{n-1}|\hat{\boldsymbol{x}}_n = \boldsymbol{x}_t) = q_{\hat{\boldsymbol{\beta}}}(\hat{\boldsymbol{x}}_{n-1}; \boldsymbol{x}_t, \boldsymbol{\epsilon}_n)$ given $\boldsymbol{x}_t \sim q_{\boldsymbol{\beta}}(\boldsymbol{x}_t|\boldsymbol{x}_0)$. When $\hat{\boldsymbol{\beta}}$ is unknown but we have $\boldsymbol{x}_0 = \hat{\boldsymbol{x}}_0$ and $\hat{\alpha}_n = \alpha_t$, we can minimize the gap between the optimal lower bound $\mathcal{F}_{score}^{(n)}(\theta^*)$ and $\log p_{\theta^*}(\hat{\boldsymbol{x}}_0)$, i.e, $\log p_{\theta^*}(\hat{\boldsymbol{x}}_0) - \mathcal{F}_{score}^{(n)}(\theta^*)$, by minimizing the following objective with respect to $\hat{\beta}_n$:*

$$\mathcal{L}_{step}^{(n)}(\hat{\beta}_n; \theta^*) := D_{\mathrm{KL}}\left(p_{\theta^*}(\hat{\boldsymbol{x}}_{n-1}|\hat{\boldsymbol{x}}_n = \boldsymbol{x}_t)||q_{\hat{\beta}_n}(\hat{\boldsymbol{x}}_{n-1}|\boldsymbol{x}_0; \alpha_t)\right) \tag{46}$$

$$= \frac{\delta_t}{2(\delta_t - \hat{\beta}_n)}\left\|\boldsymbol{\epsilon}_n - \frac{\hat{\beta}_n}{\delta_t}\boldsymbol{\epsilon}_{\theta^*}\left(\alpha_t \boldsymbol{x}_0 + \sqrt{\delta_t}\boldsymbol{\epsilon}_n, \alpha_t\right)\right\|_2^2 + C, \tag{47}$$

*where*

$$\delta_t = 1 - \alpha_t^2, \quad C = \frac{1}{4}\log\frac{\delta_t}{\hat{\beta}_n} + \frac{D}{2}\left(\frac{\hat{\beta}_n}{\delta_t} - 1\right). \tag{48}$$

*Proof.* Note that $\hat{\boldsymbol{x}}_0 = \boldsymbol{x}_0$, $\hat{\alpha}_n = \alpha_t$, $\boldsymbol{x}_t = \alpha_t \boldsymbol{x}_0 + \sqrt{1 - \alpha_t^2}\boldsymbol{\epsilon}_n$ and $p_{\theta^*}(\hat{\boldsymbol{x}}_{n-1}|\hat{\boldsymbol{x}}_n = \boldsymbol{x}_t) = q_{\hat{\boldsymbol{\beta}}}(\hat{\boldsymbol{x}}_{n-1}; \boldsymbol{x}_t, \boldsymbol{\epsilon}_n)$. When $\boldsymbol{x}_0$ is given to $p_{\theta^*}$, we can express the probability as follows:

$$p_{\theta^*}(\hat{\boldsymbol{x}}_{n-1}|\hat{\boldsymbol{x}}_n = \boldsymbol{x}_t(\boldsymbol{x}_0), \hat{\boldsymbol{x}}_0 = \boldsymbol{x}_0) \tag{49}$$

$$= \int \mathcal{N}(\boldsymbol{z}; \boldsymbol{0}, \boldsymbol{I})p_{\theta^*}(\hat{\boldsymbol{x}}_{n-1}|\hat{\boldsymbol{x}}_n = \alpha_t \boldsymbol{x}_0 + \sqrt{1 - \alpha_t^2}\boldsymbol{z})d\boldsymbol{z} \tag{50}$$

$$= \int \mathcal{N}(\boldsymbol{z}; \boldsymbol{0}, \boldsymbol{I})q_{\hat{\boldsymbol{\beta}}}(\hat{\boldsymbol{x}}_{n-1}; \boldsymbol{x}_t = \alpha_t \boldsymbol{x}_0 + \sqrt{1 - \alpha_t^2}\boldsymbol{z}, \boldsymbol{\epsilon}_n = \boldsymbol{z})d\boldsymbol{z} \tag{51}$$

$$= \int \mathcal{N}(\boldsymbol{z}; \boldsymbol{0}, \boldsymbol{I})\mathcal{N}\left(\hat{\boldsymbol{x}}_{n-1}; \frac{\alpha_t \boldsymbol{x}_0 + \sqrt{1 - \alpha_t^2}\boldsymbol{z}}{\sqrt{1 - \hat{\beta}_n}} - \frac{\hat{\beta}_n}{\sqrt{(1 - \hat{\beta}_n)(1 - \hat{\alpha}_n^2)}}\boldsymbol{z}, \frac{1 - \hat{\alpha}_{n-1}^2}{1 - \hat{\alpha}_n^2}\hat{\beta}_n\boldsymbol{I}\right)d\boldsymbol{z} \tag{52}$$

[See Eq. (2) in (Frey, 1999)]

$$= \mathcal{N}\left(\hat{\boldsymbol{x}}_{n-1}; \frac{\alpha_t \boldsymbol{x}_0}{\sqrt{1 - \hat{\beta}_n}}, \left(\left(\frac{\sqrt{1 - \alpha_t^2}}{\sqrt{1 - \hat{\beta}_n}} - \frac{\hat{\beta}_n}{\sqrt{(1 - \hat{\beta}_n)(1 - \alpha_t^2)}}\right)^2 + \frac{1 - \alpha_t^2/(1 - \hat{\beta}_n)}{1 - \alpha_t^2}\hat{\beta}_n\right)\boldsymbol{I}\right) \tag{53}$$

$$= \mathcal{N}\left(\hat{\boldsymbol{x}}_{n-1}; \frac{\alpha_t \boldsymbol{x}_0}{\sqrt{1 - \hat{\beta}_n}}, \frac{1 - \alpha_t^2 - \hat{\beta}_n}{1 - \hat{\beta}_n}\boldsymbol{I}\right) =: q_{\hat{\beta}_n}(\hat{\boldsymbol{x}}_{n-1}; \boldsymbol{x}_0, \alpha_t), \tag{54}$$

where, different from $p_{\theta^*}(\hat{\boldsymbol{x}}_{n-1}|\hat{\boldsymbol{x}}_n = \boldsymbol{x}_t)$, from Eq. (49) to Eq. (50), instead of conditioning on a specific $\boldsymbol{x}_t$, when $\boldsymbol{x}_0$ is given $\boldsymbol{x}_t$ can be generated using any $\boldsymbol{z} \sim \mathcal{N}(\boldsymbol{0}, \boldsymbol{I})$.

From this, we can express the gap between $\log p_{\theta^*}(\hat{\boldsymbol{x}}_0)$ and $\mathcal{F}_{\text{score}}^{(n)}(\theta^*)$ in the following form:

$$\log p_{\theta^*}(\hat{\boldsymbol{x}}_0 = \boldsymbol{x}_0) - \mathcal{F}_{\text{score}}^{(n)}(\theta^*) \tag{55}$$

$$= \log p_{\theta^*}(\hat{\boldsymbol{x}}_0 = \boldsymbol{x}_0) - \mathbb{E}_{p_{\theta^*}(\hat{\boldsymbol{x}}_{1,n-1}|\hat{\boldsymbol{x}}_n=\boldsymbol{x}_t)} \left[ \log \frac{p_\theta(\hat{\boldsymbol{x}}_0|\hat{\boldsymbol{x}}_1) q_{\hat{\boldsymbol{\beta}}}(\hat{\boldsymbol{x}}_{n-1}; \boldsymbol{x}_t, \boldsymbol{\epsilon}_n)}{p_\theta(\hat{\boldsymbol{x}}_{n-1}|\hat{\boldsymbol{x}}_n = \boldsymbol{x}_t)} \right] \tag{56}$$

$$= \log p_{\theta^*}(\hat{\boldsymbol{x}}_0 = \boldsymbol{x}_0) - \mathbb{E}_{p_{\theta^*}(\hat{\boldsymbol{x}}_{1:n-1}|\hat{\boldsymbol{x}}_n=\boldsymbol{x}_t)} \left[ \log \frac{p_{\theta^*}(\hat{\boldsymbol{x}}_{0:n-1}|\hat{\boldsymbol{x}}_n = \boldsymbol{x}_t)}{p_{\theta^*}(\hat{\boldsymbol{x}}_{1:n-1}|\hat{\boldsymbol{x}}_n = \boldsymbol{x}_t)} \right] \tag{57}$$

$$= \mathbb{E}_{p_{\theta^*}(\hat{\boldsymbol{x}}_{1:n-1}|\hat{\boldsymbol{x}}_n=\boldsymbol{x}_t)} \left[ \log \frac{p_{\theta^*}(\hat{\boldsymbol{x}}_{1:n-1}|\hat{\boldsymbol{x}}_n = \boldsymbol{x}_t)}{p_{\theta^*}(\hat{\boldsymbol{x}}_{1:n-1}|\hat{\boldsymbol{x}}_n = \boldsymbol{x}_t, \hat{\boldsymbol{x}}_0 = \boldsymbol{x}_0)} \right] \tag{58}$$

$$= \mathbb{E}_{p_{\theta^*}(\hat{\boldsymbol{x}}_{n-1}|\hat{\boldsymbol{x}}_n=\boldsymbol{x}_t)} \left[ \log \frac{p_{\theta^*}(\hat{\boldsymbol{x}}_{n-1}|\hat{\boldsymbol{x}}_n = \boldsymbol{x}_t)}{q_{\hat{\beta}_n}(\hat{\boldsymbol{x}}_{n-1}; \boldsymbol{x}_0, \alpha_t)} \right] \tag{59}$$

$$= D_{\text{KL}} \left( p_{\theta^*}(\hat{\boldsymbol{x}}_{n-1}|\hat{\boldsymbol{x}}_n = \boldsymbol{x}_t) \| q_{\hat{\beta}_n}(\hat{\boldsymbol{x}}_{n-1}; \boldsymbol{x}_0, \alpha_t) \right) \tag{60}$$

Next, we evaluate the above KL divergence term. By definition, we have

$$p_{\theta^*}(\hat{\boldsymbol{x}}_{n-1}|\hat{\boldsymbol{x}}_n = \boldsymbol{x}_t) = \mathcal{N} \left( \frac{1}{\sqrt{1 - \hat{\beta}_n}} \left( \boldsymbol{x}_t - \frac{\hat{\beta}_n}{\sqrt{1 - \hat{\alpha}_n^2}} \boldsymbol{\epsilon}_{\theta^*}(\boldsymbol{x}_t, \hat{\alpha}_n) \right), \frac{1 - \hat{\alpha}_{n-1}^2}{1 - \hat{\alpha}_n^2} \hat{\beta}_n \boldsymbol{I} \right) \tag{61}$$

Together with Eq. (54), we have

$$\mathcal{L}_{\text{step}}^{(n)}(\hat{\beta}_n; \theta^*) := D_{\text{KL}} \left( p_{\theta^*}(\hat{\boldsymbol{x}}_{n-1}|\hat{\boldsymbol{x}}_n = \boldsymbol{x}_t) \| q_{\hat{\beta}_n}(\hat{\boldsymbol{x}}_{n-1}; \boldsymbol{x}_0, \alpha_t) \right) \tag{62}$$

$$= \frac{1 - \hat{\beta}_n}{2(1 - \hat{\beta}_n - \alpha_t^2)} \left\| \frac{\alpha_t}{\sqrt{1 - \hat{\beta}_n}} \boldsymbol{x}_0 - \frac{1}{\sqrt{1 - \hat{\beta}_n}} \left( \boldsymbol{x}_t - \frac{\hat{\beta}_n}{\sqrt{1 - \alpha_t^2}} \boldsymbol{\epsilon}_{\theta^*}(\boldsymbol{x}_t, \alpha_t) \right) \right\|_2^2 + C \tag{63}$$

$$= \frac{1 - \hat{\beta}_n}{2(1 - \hat{\beta}_n - \alpha_t^2)} \left\| \frac{\alpha_t}{\sqrt{1 - \hat{\beta}_n}} \boldsymbol{x}_0 - \frac{1}{\sqrt{1 - \hat{\beta}_n}} \left( \alpha_t \boldsymbol{x}_0 + \sqrt{1 - \alpha_t^2} \boldsymbol{\epsilon}_n - \frac{\hat{\beta}_n}{\sqrt{1 - \alpha_t^2}} \boldsymbol{\epsilon}_{\theta^*}(\boldsymbol{x}_t, \alpha_t) \right) \right\|_2^2 + C \tag{64}$$

$$= \frac{1 - \hat{\beta}_n}{2(1 - \hat{\beta}_n - \alpha_t^2)} \left\| \sqrt{\frac{1 - \alpha_t^2}{1 - \beta_n}} \boldsymbol{\epsilon}_n - \frac{\hat{\beta}_n}{\sqrt{(1 - \beta_n)(1 - \alpha_t^2)}} \boldsymbol{\epsilon}_{\theta^*}(\boldsymbol{x}_t, \alpha_t) \right\|_2^2 + C \tag{65}$$

$$= \frac{1 - \alpha_t^2}{2(1 - \hat{\beta}_n - \alpha_t^2)} \left\| \boldsymbol{\epsilon}_n - \frac{\hat{\beta}_n}{1 - \alpha_t^2} \boldsymbol{\epsilon}_{\theta^*}(\boldsymbol{x}_t, \alpha_t) \right\|_2^2 + C \tag{66}$$

$$= \frac{\delta_t}{2(\delta_t - \hat{\beta}_n)} \left\| \boldsymbol{\epsilon}_n - \frac{\hat{\beta}_n}{\delta_t} \boldsymbol{\epsilon}_{\theta^*}\left(\alpha_t \boldsymbol{x}_0 + \sqrt{\delta_t} \boldsymbol{\epsilon}_n, \alpha_t \right) \right\|_2^2 + C, \tag{67}$$

where

$$\delta_t = 1 - \alpha_t^2, \quad C = \frac{1}{4} \log \frac{\delta_t}{\hat{\beta}_n} + \frac{D}{2} \left( \frac{\hat{\beta}_n}{\delta_t} - 1 \right). \tag{68}$$

$\square$

As we use a schedule network $\phi$ to estimate $\hat{\beta}_n$ from $(\hat{\alpha}_{n+1}, \hat{\beta}_{n+1})$ as defined in Eq. (13), we obtain the final step loss for learning $\phi$:

$$\mathcal{L}_{\text{step}}^{(n)}(\phi; \theta^*) = \frac{\delta_t}{2(\delta_t - \hat{\beta}_n(\phi))} \left\| \boldsymbol{\epsilon}_n - \frac{\hat{\beta}_n(\phi)}{\delta_t} \boldsymbol{\epsilon}_{\theta^*}(\boldsymbol{x}_t, \alpha_t) \right\|_2^2 + \frac{1}{4} \log \frac{\delta_t}{\hat{\beta}_n(\phi)} + \frac{D}{2} \left( \frac{\hat{\beta}_n(\phi)}{\delta_t} - 1 \right). \tag{69}$$

This proposed objective for training the schedule network can be interpreted as to better model the data distribution (i.e., maximizing $\log p_\theta(\hat{\boldsymbol{x}}_0)$) by correcting the gradient scale $\hat{\beta}_n$ for the next reverse step (from $\hat{\boldsymbol{x}}_n$ to $\hat{\boldsymbol{x}}_{n-1}$) given the gradient vector $\boldsymbol{\epsilon}_{\theta*}$ estimated by the score network $\theta^*$.

## B  EXPERIMENTAL DETAILS

### B.1  CONVENTIONAL GRID SEARCH ALGORITHM FOR DDPMS

We reproduced the grid search algorithm in (Chen et al., 2020), in which a 6-step noise schedule was searched. In our paper, we generalized the grid search algorithm by similarly sweeping the $N$-step noise schedule over the following possibilities with a bin width $M = 9$:

$$\{1, 2, 3, 4, 5, 6, 7, 8, 9\} \otimes \{10^{-6 \cdot N/N}, 10^{-6 \cdot (N-1)/N}, ..., 10^{-6 \cdot 1/N}\}, \tag{70}$$

where $\otimes$ denotes the cartesian product applied on two sets. LS-MSE was used as a metric to select the solution during the search. When $N = 6$, we resemble the GS algorithm in (Chen et al., 2020). Note that above searching method normally does not scale up to $N > 6$ steps for its exponential computational cost $\mathcal{O}(9^N)$.

### B.2  HYPERPARAMETER SETTING IN BDDMS

Algorithm 2 took a skip factor $\tau$ to control the stride for training the schedule network. The value of $\tau$ would affect the coverage of step sizes when training the schedule network, hence affecting the predicted number of steps $N$ for inference – the higher $\tau$ is, the shorter the predicted inference schedule tends to be. We set $\tau = 66$ for training the BDDM vocoders in this paper.

For initializing Algorithm 3 for noise scheduling, we could take as few as 1 training sample for validation, perform a grid search on the hyperparameters $\{(\hat{\alpha}_N = 0.1\alpha_T i, \hat{\beta}_N = 0.1j)\}$ for $i, j = 1, ..., 9$, i.e., 81 possibilities in total, and use the PESQ measure as the selection metric. Then, the predicted noise schedule corresponding to the maximum PESQ was stored and applied to the online inference afterward, as shown in Algorithm 4. Note that this searching has a complexity of only $\mathcal{O}(M^2)$ (e.g., $M = 9$ in this case), which is much more efficient than $\mathcal{O}(M^N)$ in the conventional grid search algorithm in (Chen et al., 2020), as discussed in Section B.1.

### B.3  IMPLEMENTATION DETAILS

Our proposed BDDMs and the baseline methods were all implemented with the Pytorch library. The score networks for the LJ and VCTK speech datasets were trained from scratch on a single NVIDIA Tesla P40 GPU with batch size 32 for about 1M steps, which took about 3 days.

For the model architecture, we used the same architecture as in DiffWave (Kong et al., 2021) for the score network with 128 residual channels; we adopted a lightweight GALR network (Lam et al., 2021) for the schedule network. GALR was originally proposed for speech enhancement, so we considered it well suited for predicting the noise scales. For the configuration of the GALR network, we used a window length of 8 samples for encoding, a segment size of 64 for segmentation and only two GALR blocks of 128 hidden dimensions, and other settings were inherited from (Lam et al., 2021). To make the schedule network output with a proper range and dimension, we applied a sigmoid function to the last block's output of the GALR network. Then the result was averaged over the segments and the feature dimensions to obtain the predicted ratio: $\sigma_\phi(\boldsymbol{x}) = \text{AvgPool2D}(\sigma(\text{GALR}(\boldsymbol{x})))$, where $\text{GALR}(\cdot)$ denotes the GALR network, $\text{AvgPool2D}(\cdot)$ denotes the average pooling operation applied to the segments and the feature dimensions, and $\sigma(x) := 1/(1 + e^{-x})$. The same network architecture was used for the NE approach for estimating $\alpha_t^2$ and was shown better than the ConvTASNet used in the original paper (San-Roman et al., 2021). It is also notable that the computational cost of a schedule network is indeed fractional compared to the cost of a score network, as predicting a noise scalar variable is intrinsically a relatively much easier task. Our GALR-based schedule network, while being able to produce stable and reliable results, was about 3.6 times faster than the score network. The training of schedule networks for BDDMs took only 10k steps to converge, which consumed no more than an hour on a single GPU.

Table 3: Ratings that have been used in evaluation of speech naturalness of synthetic samples.

| Rating | Naturalness | Definition |
|---|---|---|
| 1 | Unsatisfactory | Very annoying, distortion is objectionable. |
| 2 | Poor | Annoying distortion, but not objectionable. |
| 3 | Fair | Perceptible distortion, slightly annoying. |
| 4 | Good | Slight perceptible level of distortion, but not annoying. |
| 5 | Excellent | Imperceptible level of distortion. |

Table 4: Performances of different noise schedules on the multi-speaker VCTK speech dataset, each of which used the same score network (Chen et al., 2020) $\epsilon_\theta(\cdot)$ that was trained on VCTK for about 1M iterations.

| Noise schedule | LS-MSE ($\downarrow$) | MCD ($\downarrow$) | STOI ($\uparrow$) | PESQ ($\uparrow$) | MOS ($\uparrow$) |
|---|---|---|---|---|---|
| **DDPM (Ho et al., 2020; Chen et al., 2020)** | | | | | |
| 8 steps (Grid Search) | 101 | **2.09** | **0.787** | **3.31** | **4.22 $\pm$ 0.04** |
| 1,000 steps (Linear) | 85.0 | 2.02 | 0.798 | 3.39 | 4.40 $\pm$ 0.05 |
| **DDIM (Song et al., 2021)** | | | | | |
| 8 steps (Linear) | 553 | 3.20 | 0.701 | 2.81 | 3.83 $\pm$ 0.04 |
| 16 steps (Linear) | 412 | 2.90 | 0.724 | 3.04 | 3.88 $\pm$ 0.05 |
| 21 steps (Linear) | 355 | 2.79 | 0.739 | 3.12 | 4.12 $\pm$ 0.05 |
| 100 steps (Linear) | 259 | 2.58 | 0.759 | 3.30 | 4.27 $\pm$ 0.04 |
| **NE (San-Roman et al., 2021)** | | | | | |
| 8 steps (Linear) | 208 | 2.54 | 0.740 | 3.10 | 4.18 $\pm$ 0.04 |
| 16 steps (Linear) | 183 | 2.53 | 0.742 | 3.20 | 4.26 $\pm$ 0.04 |
| 21 steps (Linear) | 852 | 3.57 | 0.699 | 2.66 | 3.70 $\pm$ 0.03 |
| **BDDM $(\hat{\alpha}_N, \hat{\beta}_N)$** | | | | | |
| 8 steps $(0.2, 0.9)$ | **98.4** | 2.11 | 0.774 | 3.18 | 4.20 $\pm$ 0.04 |
| 16 steps $(0.5, 0.5)$ | **73.6** | **1.93** | **0.813** | **3.39** | **4.35 $\pm$ 0.05** |
| 21 steps $(0.5, 0.1)$ | **76.5** | **1.83** | **0.827** | **3.43** | **4.48 $\pm$ 0.06** |

Regarding the image generation task, to demonstrate the generalizability of our method, we directly adopted a score network pre-trained on the CIFAR-10 dataset implemented by a third-party open-source repository. Regarding the schedule network, to demonstrate that it does not have to use specialized architecture, we replaced GALR by the VGG11 (Simonyan & Zisserman, 2014), which was also used by as a noise estimator in (San-Roman et al., 2021). The output dimension (number of classes) of VGG11 was set to 1. Similar to the setting for GALR in speech synthesis, we added a sigmoid activation to the last layer to ensure a $[0, 1]$ output. Similar to the training in speech domain, we trained the VGG11-based schedule networks while freezing the score networks for 10k steps, which normally can be finished in about two hours.

Our code for the speech vocoding and the image generation experiments will be uploaded to Github after the final decision of ICLR is released.

### B.4 CROWD-SOURCED SUBJECTIVE EVALUATION

All our Mean Opinion Score (MOS) tests were crowd-sourced. We refer to the MOS scores in (Protasio Ribeiro et al., 2011), and the scoring criteria have been included in Table 3 for completeness. The samples were presented and rated one at a time by the testers.

## C ADDITIONAL EXPERIMENTS

A demonstration page at `https://bilateral-denoising-diffusion-model.github.io` shows some samples generated by BDDMs trained on LJ speech and VCTK datasets.

## C.1 MULTI-SPEAKER SPEECH SYNTHESIS

In addition to the single-speaker speech synthesis, we evaluated BDDMs on the multi-speaker speech synthesis benchmark VCTK (Yamagishi et al., 2019). VCTK consists of utterances sampled at $48$ KHz by $108$ native English speakers with various accents. We split the VCTK dataset for training and testing: 100 speakers were used for training the multi-speaker model and 8 speakers for testing. We trained on a 44257-utterance subset (40 hours) and evaluated on a held-out 100-utterance subset. For the score network, we used the Wavegrad architecture (Chen et al., 2020) so as to examine whether the superiority of BDDMs remains in a different dataset and with a different score network architecture.

Results are presented in Table 4. For this multi-speaker VCTK dataset, we obtained consistent observations with that for the single-speaker LJ dataset presented in the main paper. Again, the proposed BDDM with only 16 or 21 steps outperformed the DDPM with 1,000 steps. To the best of our knowledge, ours was the first work that reported this degree of superior. When reducing to 8 steps, BDDM obtained performance on par with (except for a worse PESQ) the costly grid-searched 8 steps (which were unscalable to more steps) in DDPM. For NE, we could again observe a degradation from its 16 steps to 21 steps, indicating the instability of NE for the VCTK dataset likewise. In contrast, BDDM gave continuously improved performance while increasing the step number.

## C.2 COMPARING DIFFERENT REVERSE PROCESSES FOR BDDMS

This section demonstrates that BDDMs do not restrict the sampling procedure to a specialized reverse process in Algorithm 4. In particular, we evaluated different reverse processes, including that of DDPMs as shown in Eq. (4) and DDIMs (Song et al., 2021), for BDDMs and compared the objective scores on the generated samples. DDIMs (Song et al., 2021) formulate a non-Markovian generative process that accelerates the inference while keeping the same training procedure as DDPMs. The original generative process in Eq. (4) in DDPMs is modified into

$$p_\theta^{(\tau)}(\boldsymbol{x}_{0:T}) := \pi(\boldsymbol{x}_T) \prod_{i=1}^{S} p_\theta^{(\gamma_i)}(\boldsymbol{x}_{\gamma_{i-1}}|\boldsymbol{x}_{\gamma_i}) \times \prod_{t \in \bar{\gamma}} p_\theta^{(t)}(\boldsymbol{x}_0|\boldsymbol{x}_t), \tag{71}$$

where $\boldsymbol{\gamma}$ is a sub-sequence of length $N$ of $[1,...,T]$ with $\gamma_N = T$, and $\bar{\gamma} := \{1,...,T\} \setminus \boldsymbol{\gamma}$ is defined as its complement; Therefore, only part of the models are used in the sampling process.

To achieve the above, DDIMs defined a prediction function $f_\theta^{(t)}(\boldsymbol{x}_t)$ that depends on $\boldsymbol{\epsilon}_\theta$ to predict the observation $\boldsymbol{x}_0$ given $\boldsymbol{x}_t$ directly:

$$f_\theta^{(t)}(\boldsymbol{x}_t) := \frac{1}{\alpha_t}\left(\boldsymbol{x}_t - \sqrt{1 - \alpha_t^2}\boldsymbol{\epsilon}_\theta(\boldsymbol{x}_t, \alpha_t)\right). \tag{72}$$

By leveraging this prediction function, the conditionals in Eq. (71) are formulated as

$$p_\theta^{(\gamma_i)}(\boldsymbol{x}_{\gamma_{i-1}}|\boldsymbol{x}_{\gamma_i}) = \mathcal{N}\left(\frac{\alpha_{\gamma_{i-1}}}{\alpha_{\gamma_i}}\left(\boldsymbol{x}_{\gamma_i} - \varsigma\boldsymbol{\epsilon}_\theta(\boldsymbol{x}_{\gamma_i}, \alpha_{\gamma_i})\right), \sigma_{\gamma_i}^2\boldsymbol{I}\right) \quad \text{if } i \in [N], i > 1 \tag{73}$$

$$p_\theta^{(t)}(\boldsymbol{x}_0|\boldsymbol{x}_t) = \mathcal{N}(f_\theta^{(t)}(\boldsymbol{x}_t), \sigma_t^2\boldsymbol{I}) \quad \text{otherwise}, \tag{74}$$

where the detailed derivation of $\sigma_t$ and $\varsigma$ can be referred to (Song et al., 2021). In the original DDIMs, the accelerated reverse process produces samples over the subsequence of $\boldsymbol{\beta}$ indexed by $\boldsymbol{\gamma}$: $\hat{\boldsymbol{\beta}} = \{\beta_n | n \in \boldsymbol{\gamma}\}$. In BDDMs, to apply the DDIM reverse process, we use the $\hat{\boldsymbol{\beta}}$ predicted by the schedule network in place of a subsequence of the training schedule $\boldsymbol{\beta}$.

Finally. the objective scores are given in Table 5. Note that the subjective evaluation (MOS) is omitted here since the other assessments above have shown that the MOS scores are highly correlated with the objective measures, including STOI and PESQ. They indicate that applying BDDMs to either DDPM or DDIM reverse process leads to comparable and competitive results. Meanwhile, the results show some subtle differences: BDDMs over a DDPM reverse process gave slightly better samples in terms of signal error and consistency metrics (i.e., LS-MSE and MCD), while BDDM over a DDIM reverse process tended to generate better samples in terms of intelligibility and perceptual metrics (i.e., STOI and PESQ).

Table 5: Performances of different reverse processes for BDDMs on the VCTK speech dataset, each of which used the same score network (Chen et al., 2020) $\epsilon_\theta(\cdot)$ and the same noise schedule.

| Noise schedule | LS-MSE ($\downarrow$) | MCD ($\downarrow$) | STOI ($\uparrow$) | PESQ ($\uparrow$) |
|---|---|---|---|---|
| **BDDM (DDPM reverse process)** | | | | |
| 8 steps $(0.3, 0.9, 1e^{-5})$ | **91.3** | **2.19** | 0.936 | 3.22 |
| 16 steps $(0.7, 0.1, 1e^{-6})$ | **73.3** | **1.88** | 0.949 | 3.32 |
| 21 steps $(0.5, 0.1, 1e^{-6})$ | **72.2** | **1.91** | 0.950 | 3.33 |
| **BDDM (DDIM reverse process)** | | | | |
| 8 steps $(0.3, 0.9, 1e^{-5})$ | 91.8 | **2.19** | **0.938** | **3.26** |
| 16 steps $(0.7, 0.1, 1e^{-6})$ | 77.7 | 1.96 | **0.953** | **3.37** |
| 21 steps $(0.5, 0.1, 1e^{-6})$ | 77.6 | 1.96 | **0.954** | **3.39** |

Table 6: Comparing sampling methods for DDPM with different number of sampling steps in terms of FIDs in CIFAR10.

| Sampling method | Sampling steps | FID |
|---|---|---|
| DDPM (baseline) (Ho et al., 2020) | 1000 | 3.17 |
| DDPM (sub-VP) (Song et al., 2020b) | $\sim 100$ | 3.69 |
| DDPM (DP + reweighting) (Watson et al., 2021) | 128
64 | 5.24
6.74 |
| DDIM (quadratic) (Song et al., 2021) | 100
50 | 4.16
4.67 |
| FastDPM (approx. STEP) (Kong & Ping, 2021) | 100
50 | 2.86
3.20 |
| [2]Improved DDPM (hybrid) (Nichol & Dhariwal, 2021) | 100
50 | 4.63
5.09 |
| VDM (augmented) (Kingma et al., 2021) | 1000 | 7.41[3] |
| Ours BDDM | 100
50 | **2.38**
**2.93** |

## C.3 UNCONDITIONAL IMAGE GENERATION

For the unconditional image generation task, we evaluated the proposed BDDMs on the benchmark CIFAR-10 ($32 \times 32$) dataset. The score functions, including those initially proposed in DDPMs (Ho et al., 2020) or DDIMs (Song et al., 2021) and those pre-trained in the above third-party implementations, are all conditioned on a discrete step-index. We estimated the noise schedule $\hat{\boldsymbol{\beta}}$ in continuous space using the VGG11 schedule network and then mapped it to discrete time schedule using the approximation method in (Kong & Ping, 2021).

Table 6 shows the performances of different sampling methods for DDPMs in CIFAR-10. By setting the maximum number of sampling steps ($N$) for noise scheduling, we can fairly compare the improvements achieved by BDDMs against related methods in the literature in terms of FID. Remarkably, BDDMs with 100 sampling steps not only surpassed the 1000-step DDPM baseline, but also produced the SOTA FID performance amongst all generative models using less than or equal to 100 sampling steps.

---

[2]Our implementation was based on `https://github.com/openai/improved-diffusion`

[3]The authors of VDM claimed that they tuned the hyperparameters only for minimizing the likelihood and did not pursue further tuning of the model to improve FID.

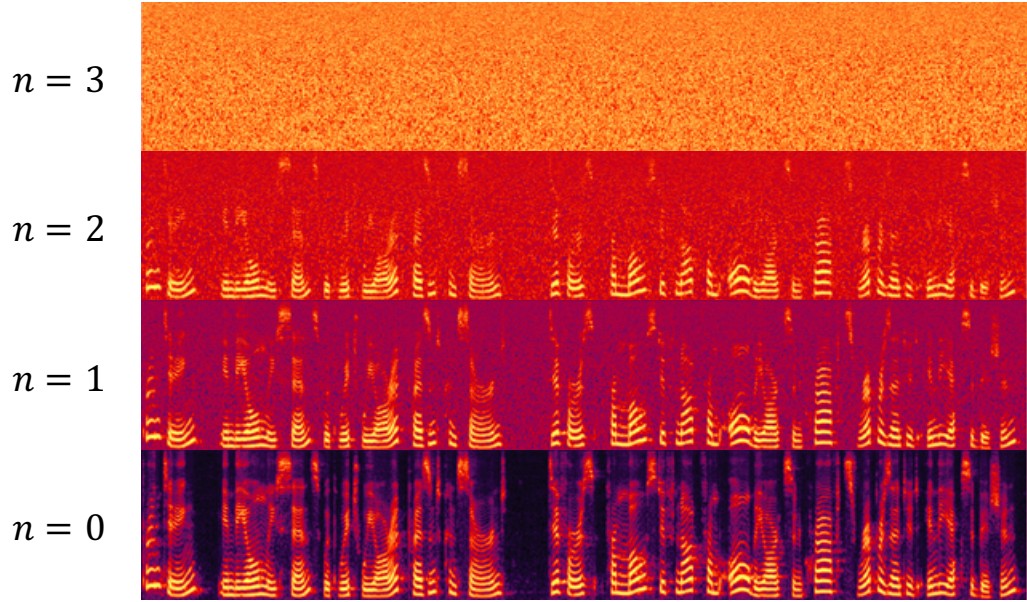

Figure 4: Spectrum plots of the speech samples produced by BDDM within 3 sampling steps. The first row shows the spectrum of a random signal for starting the reverse process. Then, from the top to the bottom, we show the spectrum of the resultant signal after each step of the reverse process performed by the BDDM. We also provide the corresponding WAV files on our demo page.

