# OpenReview forum: "BDDM: Bilateral Denoising Diffusion Models for Fast and High-Quality Speech Synthesis"
_ICLR.cc/2022/Conference — ICLR 2022 Poster_

### Official Review · Reviewer_8FYb · 2021-10-25

**Correctness:** 3
**Technical Novelty And Significance:** 3
**Empirical Novelty And Significance:** 2
**Recommendation:** 8
**Confidence:** 2

**Main Review:**

Strengths:
- Although I just did a quick pass, I think the derivations in Appendix A are correct.
- Results in Table 1 seem convincing.
- Results for VCTK are nice. I'd suggest adding them to the main paper.
- I found the ablation of ELBO vs BDDM appropriate.

Weaknesses:
- I had difficulty in understanding the paper. I think it is not accessible for a wide audience.
- I personally find that the paper lacks motivation in every step. High-level interpretations are missing and quite often one has the impression that the corresponding formula "just appears" in there, somehow randomly or out of nowhere. After carefully reading it for hours I still do not understand the main intuition behind the proposed approach: Why this way?
- The paper discusses and proposes generic approaches in a general setting, but then only empirically evaluates them in the task of speech vocoding, which is in a quite particular domain (speech) and, more importantly, corresponds to a strongly-conditioned task (that is, a task where the conditioning carries a lot of information; for instance, it is known that tasks like that require much less sampling steps for diffusion models). It leaves a lot of doubt of whether the proposed approach works with, say, unconditional image generation, or even a less strongly-conditioned task like category-based generation.
- I'm not convinced about the significance of the improvement between the methods presented in Tables 1 and 2 (I'm talking about perceptual significance, not statistical significance here). Furthermore, for some methods and measures (and specially in Table 2), confidence intervals are quite wide, leaving the reader with the doubt of whether the approach is worth this complicated development when compared to, for instance, DDIM.

Other/random:
- I think the role of $\tau$ is not discussed enough.
- I'd have appreciated more discussion on the role of the architecture for the scheduling network.
- Why GS becomes "prohibitively" for $N>6$ and not $N>5$ or $N>10$? Even with a reduced set of synthesis utterances?

**Summary Of The Paper:**

The paper proposes to train a scheduling network in addition to the score network for fast sampling of diffusion models. To train such network a new loss is proposed. Results are presented for the task of neural vocoding (single-speaker and multi-speaker).

**Summary Of The Review:**

I think the proposal of the paper is sound and the developments are correct. However, the choice of the task for which to provide empirical evidence is in a single domain (speech) and very specific (extremely strong conditioning). I personally find the paper hard to follow and lacking motivation/intuition.

[Update: After authors' response and improved manuscript I decided to raise my score from 6 to 8]

---

> ### Author Response · Authors · 2021-11-22
> **Thanks for your comment**
>
> We thank the reviewer for the suggestions. We understand that your concerns are mainly related to the paper's readability. We hope that our amendments to the paper resolve your doubts fully.
>
> > "I personally find that the paper lacks motivation in every step. High-level interpretations are missing ..."
>
> We find it a very beneficial suggestion for improving our paper's readability.
> - To position the novelty of our approach and present the challenge that BDDMs confront, we added a section of "Related Work" and a section about "Problem Formulation" in the revised paper.
> - Following this suggestion and those by Reviewer vo24, we have added high-level explanations and justifications for our equations to help readers better understand the problems, motivations, solutions, and why they are necessary -- Meanwhile, the original theoretical propositions are moved to Appendix A.
>
> > "The paper discusses and proposes generic approaches in a general setting, but then only empirically evaluates them in the task of speech vocoding, which is in a quite particular domain (speech) and, more importantly, corresponds to a strongly-conditioned task (that is, a task where the conditioning carries a lot of information; for instance, it is known that tasks like that require much less sampling steps for diffusion models). It leaves a lot of doubt of whether the proposed approach works with, say, unconditional image generation, or even a less strongly-conditioned task like category-based generation."
>
> Yes, BDDMs are also effective and applicable to unconditional image generation.
>
> - We conducted an additional experiment of applying BDDM to a benchmark unconditional image generation task on CIFAR-10. The result is shown in Appendix C.3 and compared to other relevant methods provided with the FID results using 50 and 100 sampling steps.
>
> - Remarkably, BDDMs with 100 sampling steps not only surpassed the 1000-step DDPM baseline but also produced an FID of **2.38**, which, to the best of our knowledge, reaches SOTA amongst all DPMs using comparable sampling steps.
>
> > "I'm not convinced about the significance of the improvement between the methods presented in Tables 1 and 2 (I'm talking about perceptual significance, not statistical significance here). Furthermore, for some methods and measures (and specially in Table 2), confidence intervals are quite wide...DDIM"
>
> - Since MOS has been commonly used as the standard measure of perceptual quality for evaluating neural vocoders, e.g., in highly-cited works [1-3], we believe that statistical significance in MOS should be highly relevant to the perceptual significance.
> - We consider the wider CIs of MOS compared to other neural vocoding papers were primarily due to fewer raters (12 listeners rated in our current report). We are enrolling more participants and promise to increase the number to 30 and update the report before submitting the final version.
> - Besides the MOS scores, the objective measures in Table 2 can also reference sample quality. We believe that the consistent and significant improvements by BDDMs in the objective measures can also support our claim of a superior fast sampling method for DPMs.
> - Moreover, our newly added results for the unconditional image generation experiments further validated the significance and superiority of the proposed BDDMs.
>
> [1] van den Oord, Aäron, et al. "WAVENET: A GENERATIVE MODEL FOR RAW AUDIO." 2016.
> [2] K., Nal, et al. "Efficient neural audio synthesis." ICML. PMLR, 2018.
> [3] O., Aaron, et al. "Parallel wavenet: Fast high-fidelity speech synthesis." ICML. PMLR, 2018.
>
> > "I think the role of $\tau$ is not discussed enough."
> - A thorough description of the role of $\tau$ was added to the beginning of Section 4.2 (starting from page 5) of the revised paper.
>
> > "I'd have appreciated more discussion on the role of the architecture for the scheduling network."
> - We separated a new Section 4.3 to describe the role of the schedule network-- predicting the noise scale from a noisy input $x\_t$, which resembles a speech separation task. Therefore, as presented in Appendix B.3, we constructed the scheduling network with a GALR architecture originally designed for speech separation.
> - Meanwhile, we empirically found learning a noise schedule a much easier task than learning a score network or a speech separation network. Therefore, the scheduling network was more light-weighted with fewer layers and converged much faster.
>
> > "Why GS becomes "prohibitively" for $N>6$ and not $N>5$ or $N>10$? Even with a reduced set of synthesis utterances?"
> - In our experiment, $N=6$ took more than a day on a single NVIDIA Tesla P40 GPU (even using only one training example). This is because the time costs of GS algorithm grow exponentially with $N$, i.e., $\mathcal{O}(9^N)$ with 9 bins as the default setting in WaveGrad.
> - This explanation is also added to the Section 2 of the revised paper.

---

> > ### Comment · Reviewer_8FYb · 2021-11-26
> > **Thanks**
> >
> > Thank you for your answer improved version of the paper. After this, I think the paper deserves a chance and therefore raise my score from 6 to 8. I hope that the authors can polish a bit more the presentation as suggested by other reviewers in case of acceptance.

---

### Official Review · Reviewer_LPmT · 2021-11-01

**Correctness:** 3
**Technical Novelty And Significance:** 3
**Empirical Novelty And Significance:** 3
**Recommendation:** 6
**Confidence:** 4

**Main Review:**

**Strengths**:
- The idea proposed by the paper, this is, learning a noise scheduling network based on an adapted objective such that learning this network corresponds to explicitly tightening the lower bound on the data log-likelihood, is new to the best of my knowledge (although related methods exist). Since the noise schedule is learnt specifically for synthesis with fewer steps, the work addresses an important problem of denoising diffusion models, their slow sampling speed.
- The experimental results are promising. The method achieves almost state-of-the-art results, while requiring much fewer synthesis steps, and also seems to be superior to several previous approaches for accelerating sampling in these types of models.

**Weaknesses and Questions/Suggestions**:

Overall, I found the paper difficult to follow and lacking in clarity. I have several questions and concerns:
- While the overall approach seems to be novel, it is closely related to [1]. [1] also learns the noise schedule and additionally also shows how one can perform synthesis with fewer steps. I think this suggests a more thorough, quantitative comparison to [1]. In particular, it would be interesting to learn the noise schedule in a similar manner as [1] and then also reduce the number of steps similarly ([1] works on images, but the method is general). I would be curious which method performs better. The current ablation is a bit inconclusive in that regard, as [1] uses a significantly different noise parametrization, which may not as easily collapse.
- The paper has no thorough Related Work section, which has become a standard part of many papers to appropriately position the work in the literature. I would recommend including such a section in the paper.
- Beginning of section 3; "...which is used to reversely sample $x_n \sim q_\phi(x_n|x_{n+1}, x_0)$ using a scheduling network $\phi$.": Why do we even need an additional scheduling network here? When conditioning on both $x_{n+1}$ and $x_0$, as indicated in the distribution, the ideal denoising distribution, or posterior, is tractable [2].
- In Figure 1, both $q_\phi(x_n|x_{n-1})$ and $q_\phi(x_n|x_{n+1}, x_0)$ have $\phi$ indices and depend on the learnt scheduling network. However, these are different distributions (one defines a forward process, the other the backward posterior). It is unclear, which distribution exactly is learnt here and how the forward and backward processes depend on the learnt distribution. I would suggest to clarify.
- Below Proposition 2; "In this regard, the proposed lower bound $F^{(t)}_{score}(\theta)$ allows us to consider only one $t$ at each training step for efficient training, which is practically more advantageous than the standard ELBO in Eq. (1) that entails computing a sum of T KL terms." -> In DDPM and related methods, we never actually calculate the whole sum either. Rather, $t$ is randomly sampled within a mini batch. Furthermore, my understanding is that BDDM in fact uses a DDPM objective to learn the score network as well. Hence, the advantage is not really clear to me.
- Above Proposition 2, $\pi(x_t)$ is defined as the "prior" at $t$, which, I think, we can interpret as the marginal distribution over all possible $x_t$ at $t$. But now in Proposition 3, we assume $\pi(x_{t-1})=q_\beta(x_{t−1}|x_t, x_0)$, which corresponds to the posterior over $x_{t−1}$, given specific conditionings $x_t$ and $x_0$. This seems very different compared to the $\pi(x_t)$ described above Prop. 2 and might even be contradictory. I think this needs clarification.
- What specifically is $p_{\theta^{*}}(x_{1:t−1}|x_0)$ in Proposition 3? My understanding is that $p_{\theta}$ generally is the learnt score model that defines a distribution of the form $p_\theta(x_{t−1}|x_t)$. How does this distribution then predict $x_{1:t−1}$ given $x_0$, as indicated?
- Theorem 1: In practice, it is highly unlikely that we ever have $\theta=\theta^*$. Hence, I believe the $\theta\neq\theta^*$ case is the only practically relevant one. But in this case, it isn't clear to me why the bound will be tighter than the regular ELBO (because the $\mathcal{L}_{step}$ term isn't there). The proof says it follows from Proposition 2, but it isn't clear. The proposition does not explicitly compare the derived bound with the regular ELBO bound from DDPMs in terms of how tight the bounds are, unless I am missing something.
- Eq. 19 is supposed to define a linear noise schedule, but has $t$ in the denominator. Is there an error?
- [3] also learns a noise schedule. I believe it also deserves to be discussed and cited.

[1] Nichol and Dhariwal, "Improved Denoising Diffusion Probabilistic Models", 2021.

[2] Ho et al., "Denoising Diffusion Probabilistic Models",  2020.

[3] Kingma et al. "Variational Diffusion Models", 2021.

**Summary Of The Paper:**

The paper aims at improving generative denoising diffusion models for audio synthesis. In particular, it proposes a method to learn a noise prediction or scheduling network that optimally chooses the noise scales (this is, the scales of the noise that is injected during the iterative synthesis process) at inference time. This allows for faster synthesis without much reduction in quality. The method, called Bilateral Denoising Diffusion Model (BDDM), relies on a slightly modified training objective compared to previous works, in which case learning the noise schedule corresponds to optimizing the objective towards a tighter bound on the data log-likelihood. The main score model is learned in a similar fashion like previous diffusion models, such that the objectives for the score model and the noise scheduling network decompose into two separate objectives that can be optimized essentially one after another. Experimentally, BDDM achieves audio generation with a quality almost as good as the current state-of-the-art, while requiring much less synthesis steps. It also outperforms other methods for accelerated sampling from diffusion models for audio synthesis.

**Summary Of The Review:**

Overall, the paper presents a new idea for learning the inference noise schedule in generative diffusion models and the experimental results are promising. However, closely related methods exist [1] and a more detailed comparison isn't presented (it's not one of the baselines). My main concern is that I found the presentation and writing of the paper lacking in clarity and difficult to follow, as indicated by my questions above. This also makes it difficult to check the derivations and develop a solid intuition for the method. Related to this, the motivation for the specific approach and many steps in the derivations are a bit unclear. In conclusion, I think the paper is not ready for publication in its current form. However, the method does seem promising nonetheless. Hence, I would be willing to raise my score if the weaknesses and questions can be addressed.

---

> ### Author Response · Authors · 2021-11-23
> **(2/2) Thanks for your comment**
>
>
> > "Below Proposition 2; "In this regard, the proposed lower bound $\\mathcal{F}\_\text{score}^{(t)}$ allows us to consider only one $t$ at each training step for efficient training, which is practically more advantageous than the standard ELBO in Eq. (1) that entails computing a sum of T KL terms." -> In DDPM and related methods, we never actually calculate the whole sum either. Rather, $t$ is randomly sampled within a mini batch. Furthermore, my understanding is that BDDM in fact uses a DDPM objective to learn the score network as well. Hence, the advantage is not really clear to me."
>
> - We agree that summing over $T$ terms was only theoretically derived for ELBO but not used in practice in related works. Thereby we have removed this advantage statement in the revised paper.
> - Instead, we attributed our advantage mainly to the schedule network. We also stressed its theoretical and practical connection with the score network. Please check out the revised statement in Section 4.4, the last part of Section 4.3, and Section 6.2.
> - Yes, the reviewer's understanding is correct about the reusable DDPM score network for BDDMs. For tasks like image generation that require many days of score-network training, this property is especially convenient for us developers to conduct prompt experiments since we can directly load the 3rd-party pretrained checkpoints (see links in the paper).
>
> > "Above Proposition 2, $\pi(x\_t)$ is defined as the "prior" at $t$, which, I think, we can interpret as the marginal distribution over all possible $x_t$ at $t$. But now in Proposition 3, we assume $\pi(x\_t)=q\_\beta(x\_{t-1}|x\_t, x\_0)$, which corresponds to the posterior over $x\_{t-1}$, given specific conditionings $x\_{t}$ and $x\_{0}$. This seems very different compared to the $\pi(x\_t)$ described above Prop. 2 and might even be contradictory. I think this needs clarification."
>
> - No, we do not mean to interpret the prior as the marginal distribution. When considering a reverse process of $n$ steps with $\hat{x}_n$ being the beginning variable, $\pi(\hat{x}_n)$ was defined as the prior instead of a marginal distribution.
> - To avoid any confusion here, we revised the relevant notations. As defined in Section 4.3, we specifically consider an informative prior $q\_{\hat{\beta}}(\hat{x}\_{n-1};{x}\_{t}, \epsilon\_n))$ (parameterized by $x_t$ and $\epsilon_n$, rather than integrated over) for starting the reverse process.
>   - Notably, according to Eq. (1) in DDPM[4], $x_T\sim p(x_T)$ with $p$ being an user-defined prior that theoretically does not limited to a white noise, although the authors set $p(x_T)=\mathcal{N}(\bf{0}, \bf{I})$ in their sampling algorithm.
>
> [4] Ho, Jonathan, Ajay Jain, and Pieter Abbeel. "Denoising diffusion probabilistic models." 2020.
>
> > "What specifically is $p\_{\theta*}(x\_{1:t-1}|x\_0)$ in Proposition 3? My understanding is that $p\_{\theta}$ generally is the learnt score model that defines a distribution of the form $p\_{\theta*}(x\_{1:t-1}|x\_0)$. How does this distribution then predict $x\_{1:t-1}$ given $x\_0$, as indicated?"
>
> - Based on the revised notations, $\theta\approx\theta^{*}$ is the solution for minimizing $\mathbb{KL}(p\_{\theta}(\hat{x}\_{n-1}|\hat{x}\_{n})||q\_{\hat{\beta}}(\hat{x}\_{n-1};{x}\_{t}, \epsilon\_n))$.
> - We therefore revised the relevant condition of Proposition 3 to $p\_{\theta^{*}}(\hat{x}\_{n-1}|\hat{x}\_{n})=q\_{\hat{\beta}}(\hat{x}\_{n-1};{x}_{t}, \epsilon_n)$, which is asymtopically true and can also be used to derive $\mathcal{L}_\text{step}^{(n)}$.
>
> > "Theorem 1: In practice, it is highly unlikely that we ever have $\theta=\theta^*$. Hence, I believe the $\theta\neq\theta^*$ case is the only practically relevant one. But in this case, it isn't clear to me why the bound will be tighter than the regular ELBO (because the $\mathcal{L}\_\text{step}$ term isn't there). The proof says it follows from Proposition 2, but it isn't clear. The proposition does not explicitly compare the derived bound with the regular ELBO bound from DDPMs in terms of how tight the bounds are, unless I am missing something."
>
> - The original condition for Theorem 1 appears to be unlikely, so we instead empirically validate if the values of our proposed lower bound are always higher than the regular ELBO in Section 6.2.
>
> > "Eq. 19 is supposed to define a linear noise schedule, but has $t$ in the denominator. Is there an error?"
> - Yes, we are thankful to the reviewer for pointing out a mistake here. We corrected the definition in the revised paper.
>
> > "[3] also learns a noise schedule. I believe it also deserves to be discussed and cited."
> - In the revised paper, we distinguished the noise scheduling method used in [3] in the Related work.
> - We also compared BDDMs to [3] in a benchmark image generation task, where BDDMs have excelled amongst all examined methods in terms of FID. Please refer to detailed results in Table 6 of Appendix C.3.

---

> > ### Comment · Reviewer_LPmT · 2021-11-25
> > **Thank you for rebuttal.**
> >
> > I would like to thank the authors for their detailed rebuttal. Several of my questions have been addressed. I also think that the further experiments are insightful and support the paper. I still think that the paper is very hard to read and follow, although some helpful improvements have been made and I indeed understand the approach better now.
> >
> > Apart from the presentation, although the paper feels a bit incremental, the method itself is novel and seems useful and interesting to the community. In conclusion, I am now leaning towards accepting the paper (I raised my score from 5 to 6), while encouraging the authors to further polish the presentation for the final version in case of acceptance.

---

> ### Author Response · Authors · 2021-11-23
> **(1/2) Thanks for your comment**
>
> Thanks for the constructive and detailed comments! We understand that your concerns are mainly related to some specific parts. We hope that our clarifications below and appropriate amendments to the paper resolve your doubts fully.
>
> > "While the overall approach seems to be novel, it is closely related to [1]. [1] also learns the noise schedule and additionally also shows how one can perform synthesis with fewer steps. I think this suggests a more thorough, quantitative comparison to [1]. In particular, it would be interesting to learn the noise schedule in a similar manner as [1] and then also reduce the number of steps similarly ([1] works on images, but the method is general). I would be curious which method performs better. The current ablation is a bit inconclusive in that regard, as [1] uses a significantly different noise parametrization, which may not as easily collapse."
>
> - We would like to clarify that [1] does not learn the noise schedule. Instead, it proposed learnable variances only for the reverse process, which still required a pre-specified $\boldsymbol\beta$ (as stated in Section 3.1 of [1]).
>
> - For noise scheduling, [1] proposed a novel cosine noise schedule, which, still, is not learnable (see Eq. (17) in [1]).
>
> - For fast sampling, [1] proposed a rescaling technique based on a pre-specified subsequence of the time indices (see Eq. (19) in [1]), which, similar to DDIMs, assumes a linear or quadratic rule. In contrast, in BDDM, we are not limited to any pre-defined (e.g., linear or quadratic) rule.
>
> - Our paper has introduced the variance learning approach of [1], which can be referred to the description after our Eq. (4) raising the variance term of the reverse process. In fact, the learnable variance learning approach of [1] is applicable to extend BDDMs and could co-exist with our learnable noise scheduling approach.
>
> - To answer "which method performs better?"-- we also added a comparison of BDDM against [1] on unconditional image generation. As shown in Table 6 of Appendix C.3, BDDMs produced an FID of **2.38**, which, as far as we know, reaches the SOTA amongst all DPMs ( including [1] with an FID of **4.63**) that take comparable sampling steps.
>
> > "The paper has no thorough Related Work section, which has become a standard part of many papers to appropriately position the work in the literature. I would recommend including such a section in the paper."
>
> -  We have added a "Related Work" section to better position the novelty of BDDMs in the literature.
>
> >  "Beginning of section 3; "...which is used to reversely sample $x\_n \sim q\_\phi(x\_n|x\_{n+1}, x\_0)$ using a scheduling network $\phi$.": Why do we even need an additional scheduling network here? When conditioning on both $x\_n$ and $x\_0$, as indicated in the distribution, the ideal denoising distribution, or posterior, is tractable [2]."
>
> -  See a revised Section 4 for a clearer BDDM model formulation, based on which the necessity of a schedule network is clarified as follows.
> - The reviewer is right about the tractable posterior -- however, the sampling noise schedule ${\hat{\boldsymbol\beta}}$ is unknown in our problem formulation.
>   - As clarified in Section 4.1, our goal is to optimize the unknown $\hat{\boldsymbol\beta}$ for the revere process $p\_\theta(\hat{x}\_{n-1}| \hat{x}\_{n})$, given that $\theta$ is trained with the known ${\boldsymbol\beta}$.
>   - The only constraints we have about ${\hat{\boldsymbol\beta}}$ are the following: (1) $N\ll T$, and (2) a range of $\hat{\beta}_n$ as derived in Remark 1.
>   - As defined in Section 4.4, the schedule network $\phi$ can be used to parameterize $\hat{\beta}_n(\phi)$. We also prove in Appendix A.3 that $\mathcal{L}\_\text{step}^{(n)}(\phi;\theta^*)$ can minimize the gap between the data likelihood and the optimized lower bound.
> - The schedule network $\phi$, in simple words, is trained for reverting the much shorter diffusion process (of length $N$), using the score network $\theta$ that was trained on the long diffusion process (of length $T$).
>
> > "In Figure 1, both $q\_\phi(x\_n|x\_{n-1})$ and $q\_\phi(x\_n|x\_{n+1}, x\_0)$ have $\phi$ indices and depend on the learnt scheduling network. However, these are different distributions (one defines a forward process, the other the backward posterior). It is unclear, which distribution exactly is learnt here and how the forward and backward processes depend on the learnt distribution. I would suggest to clarify."
>
> Great suggestion!
> - We have updated Figure 1 accordingly
>   - Two diffusion processes, specified by ${\boldsymbol\beta}$ and ${\hat{\boldsymbol\beta}}$, respectively, are linked by a so-called junctional variable $x\_t$.
>   - The forward and backward processes against different distributions are now added, described, and marked with colors.
> - Also, we have revised Section 4.4 to clarify the role that $\phi$ plays for parameterizing $\hat{\beta}\_n$ in the forward process.

---

### Official Review · Reviewer_vo24 · 2021-11-01

**Correctness:** 4
**Technical Novelty And Significance:** 3
**Empirical Novelty And Significance:** 3
**Recommendation:** 6
**Confidence:** 3

**Main Review:**

Pros:

1. The idea to find the optimal forward process is interesting and novel (modulo concurrect work like VDM https://arxiv.org/abs/2107.00630 which attempts to reach the same goal).
2. The experimental results persuade that the method indeed works well with few inference steps.

##########################################################################

Cons:

1. There are a lot of technical details which make the paper hard to read. It would profit from moving some parts into the supplementary material. In particular, Section 3 first lists 2 pages of propositions and theorems - and only later the reader understands why those were necessary. It might be better to first state what is being done in simple language/terms, and only then explain it mathematically - or even move the math into the supplementary.
2. The comparison methodology to DiffWave is a bit strange: the paper correctly states that confidence intervals for both proposed method and DiffWave cover ground truth under t-test, but why not perform paired test to determine which method is better? I.e. show two generations and ask the rater which is more realistic. Also, there exist paired tests with less strict assumptions than t-test. For example, Wilcoxon signed-rank test doesn't assume normal distribution of scores: https://en.wikipedia.org/wiki/Wilcoxon_signed-rank_test.

##########################################################################

Questions during rebuttal period:

Please address the cons above.

#########################################################################

Minor suggestions and typos:

(1) Parallel WaveNet is worth mentioning as one of the earlier attempts to apply distillation for speeding up generation.

(2) "Bilateral" in the paper name is slightly confusing: it made me think about https://en.wikipedia.org/wiki/Bilateral_filter, which is unrelated to the proposed method.

**Summary Of The Paper:**

This speech synthesis paper proposes to learn the forward diffusion process and reach better ELBO values by doing that, as well as facilitate inference with fewer steps while keeping good generation quality. The experiments were conducted on LJSpeech and VCTK datasets. The paper compared the proposed method to some strong baselines, including WaveNet, GANs and two recent diffusion methods: WaveGrad and DiffWave. The metrics were STOI, PESQ and human evaluation MOS. The MOS results for the proposed method show no significant difference from the ground truth (same as DiffWave) starting from 7 steps while being and an order of magnitude faster than the best baseline DiffWave.

**Summary Of The Review:**

I am inclined to accept this paper, because it contains novel ideas and the experiments are solid. Writing would profit from simplification though.

---

> ### Author Response · Authors · 2021-11-22
> **Thanks for your comment**
>
> We are very grateful for your feedback and find them quite helpful for improving our paper. We added clarifications and new experiments and hope that your concerns have been addressed in full. If not, we are looking forward to further discussion. Please see each detail below.
>
> > "There are a lot of technical details which make the paper hard to read. It would profit from moving some parts into the supplementary material. In particular, Section 3 first lists 2 pages of propositions and theorems - and only later the reader understands why those were necessary. It might be better to first state what is being done in simple language/terms, and only then explain it mathematically - or even move the math into the supplementary."
>
> - For better clarity and readability, we added concise, descriptive language before math and moved the math derivations to the Appendix as suggested by the reviewer.
> - To clearly state the paper's motivation, we added a section on "problem formulation."
>
> > "The comparison methodology to DiffWave is a bit strange: the paper correctly states that confidence intervals for both proposed method and DiffWave cover ground truth under t-test, but why not perform paired test to determine which method is better? I.e. show two generations and ask the rater which is more realistic. Also, there exist paired tests with less strict assumptions than t-test. For example, Wilcoxon signed-rank test doesn't assume normal distribution of scores: https://en.wikipedia.org/wiki/Wilcoxon_signed-rank_test."
>
> - As suggested by the reviewer, we conducted an A/B paired test. The result shows no significant perceptual difference between the samples generated by BDDM (12 steps) and DiffWave (200 steps)
>   - In the A/B test, randomly permuted speech samples generated by **A**: BDDM (12 steps) and **B**: DiffWave (200 steps) were presented to listeners, who were asked to select a preferred sample with better speech naturalness or select no preference (**NP**). We collected 100 evaluations from 10 raters
>   - We report the result here: |**A**=18.3% | **B**=28.3% | **NP**=53.3% |. Finally, **NP** dominates the result, showing that the listeners perceived no significant preference.
> - The above A/B test result echos the MOS test result we reported in our original submission, where we presented that the MOS score of our BDDM (with 12 steps) shows NO statistical difference to that of DiffWave (with 200 steps), but ours took only 1/16 sampling time.

---

> > ### Comment · Reviewer_vo24 · 2021-11-27
> > **Response from reviewer**
> >
> > Thank you! I am keeping my score (6), leaning towards acceptance. In case the paper is accepted, please include A/B test results to the main text, this is quite informative for the reader.

---

### Decision · Program_Chairs · 2022-01-20

**Decision:**

Accept (Poster)

**Comment:**

This work suggests an extension of diffusion-based generative models, where both the forward and reverse process have learnable parameters (rather than just the reverse process). This is then applied to speech synthesis, with high-fidelity audio generated in very few sampling steps compared to what is typical for this class of models. The proposed model is specifically compared to other diffusion-based approaches for speech synthesis in terms of inference speed.

Reviewers highlighted the novelty of the idea and the convincing experimental results. Concerns were raised about the accessibility and clarity of the presentation (structure, too many technical details), lack of a related work section, and the methodology used to compare the proposed model against baselines. The authors have attempted to address these issues, and two reviewers raised their scores as a result. All reviewers now recommend acceptance.

I am therefore recommending acceptance as well, but I would like to encourage the authors to polish the presentation further, in order to make the work maximally accessible to a wide audience.